

# CAMELS-BR: Hydrometeorological time series and landscape attributes for 897 catchments in Brazil

Vinícius B. P. Chagas[1], Pedro L. B. Chaffe[2], Nans Addor[3], Fernando M. Fan[4], Ayan S. Fleischmann[4], Rodrigo C. D. Paiva[4], Vinícius A. Siqueira[4]

[1]Department of Sanitary and Environmental Engineering, Graduate Program of Environmental Engineering, Federal University of Santa Catarina–UFSC, Florianopolis, Brazil
[2]Department of Sanitary and Environmental Engineering, Federal University of Santa Catarina–UFSC, Florianopolis, Brazil
[3]Department of Geography, University of Exeter, Exeter, United Kingdom
[4]Hydraulic Research Institute, Federal University of Rio Grande do Sul-UFRGS, Porto Alegre, Brazil

*Correspondence to*: Pedro L. B. Chaffe (pedro.chaffe@ufsc.br)

**Abstract** We introduce a new catchment dataset for large-sample hydrological studies in Brazil. This dataset encompasses daily time series of observed streamflow from 3713 gauges, as well as meteorological forcing (precipitation, evapotranspiration and temperature) for 897 selected catchments. It also includes 63 attributes covering a range of topographic, climatic, hydrologic, land cover, geologic, soil and human intervention variables, as well as data quality indicators. This paper describes how the hydrometeorological time series and attributes were produced, their primary limitations and their main spatial features. To facilitate comparisons with catchments from other countries, the data follow the same standards as the previous CAMELS (Catchment Attributes and MEteorology for Large-sample Studies) datasets for the United States, Chile and Great Britain. CAMELS-BR complements the other CAMELS datasets by providing data for hundreds of catchments in the tropics and in the Amazon rainforest. Importantly, precipitation and evapotranspiration uncertainties are assessed using several gridded products and quantitative estimates of water consumption are provided to characterize human impacts on water resources. By extracting and combining data from these different data products and making CAMELS-BR publicly available, we aim to create new opportunities for hydrological research in Brazil and to facilitate the inclusion of Brazilian basins in continental to global large-sample studies. We envision that this dataset will enable the community to gain new insights into the drivers of hydrological behavior, better characterize extreme hydroclimatic events, and explore the impacts of climate change and human activities on water resources in Brazil. The CAMELS-BR dataset is freely available at https://doi.org/10.5281/zenodo.3709338 (Chagas et al., 2020).

## 1 Introduction

Large-scale hydrological research relies on data from large samples of catchments to formulate general conclusions on hydrological processes and models (Gupta et al., 2014; Addor et al., 2019). Hydrometeorological datasets with large spatial and temporal coverage are the basis to improve hydrological understanding with appropriate statistical robustness. For example, multiple studies used large-sample datasets to investigate the drivers of hydrological change (e.g., Slater et al., 2015;



Blöschl et al., 2019a; Gudmundsson et al., 2019), the impacts of anthropic activities on the water cycle (e.g., Milliman et al., 2008; Hoekstra and Mekonnen, 2012; Montanari et al., 2013), hydrological similarity and classification (e.g., Berghuijs et al., 2014; Sawicz et al., 2014; Knoben et al., 2018), predictions in ungauged basins (e.g., Yadav et al., 2007; Ehret et al., 2014;

Singh et al., 2014), areas where extreme events are a concern (e.g., Van Lanen et al., 2013; Villarini, 2016; Woldemeskel and Sharma, 2016), and to predict future hydrological change (e.g., Luke et al., 2017; Zscheischler et al., 2018). Moreover, large-sample hydrology is needed for validation of continental to global hydrological models, to identify limitations in model structure, parameterization and forcing according to geographic and climatic regions (Haddeland et al, 2011; Gudmundsson et al., 2012; Beck et al, 2017a; Zhao et al., 2017; Siqueira et al., 2018; Veldkamp et al., 2018), to estimate uncertainty in model

estimates (e.g., Müller Schmied et al., 2014; Beck et al., 2016; Hirpa et al., 2018) and to make use of data assimilation techniques (e.g., Wongchuig et al., 2019). Better predictions in such models allow for the quantification of water resources availability over large scales and are fundamental for nationwide water resources planning and management (Schewe et al., 2014; Bierkens, 2015; Doll et al., 2016; Alfieri et al., 2020).

To uncover the hydrological functioning of a catchment, it is key to understand how it is controlled by climate, human

interferences (Wohl et al., 2012; Di Baldassarre et al., 2018), and landscape attributes, such as vegetation, topography, soil, and lithology (Fan et al., 2019). For this, researchers must work with a multiple-hypotheses framework (Merz et al., 2012; Pfister and Kirchner, 2017), which frequently leads to processing massive amounts of data and often tedious, repetitive tasks. To deal with this problem, multiple datasets have been created, such as the Global Runoff Data Centre (GRDC, 2019), the Global Streamflow Indices and Metadata Archive (GSIM; Do et al., 2018; Gudmundsson et al., 2018); the HydroATLAS

(Linke et al., 2019); and the Global Runoff Reconstruction (GRUN; Ghiggi et al., 2019). A noteworthy dataset, the Catchment Attributes and MEteorology for Large-sample Studies (CAMELS; Newman et al., 2015; Addor et al., 2017), produced a synthesis from multiple catchment attributes. It initially included only catchments in the United States, but later expanded to include Chile with the CAMELS-CL dataset (Alvarez-Garreton et al., 2018) and Great Britain with CAMELS-GB (Coxon et al. (submitted)). The CAMELS datasets facilitated hydrological research by addressing some of the major problems with large

datasets, such as a lack of common standards across databases, absence of uncertainty estimation, and open accessibility of hydrological observations (Addor et al., 2019).

Even though there is a growing number of large-sample datasets worldwide (e.g., Addor et al., 2017; Do et al., 2018; Gudmundsson et al., 2018; Ghiggi et al., 2019; Linke et al., 2019), the access to open and readily available data in some regions like South America is still difficult and requires additional quality checks (Crochemore et al., 2019). Particularly in Brazil,

large-sample hydrological studies lack a comprehensive dataset to rely on. Brazilian hydrometeorological information is currently collected, maintained and distributed by institutions as the Brazilian National Water Agency (ANA – *Agência Nacional de Águas*; http://snirh.gov.br/hidroweb) and the National Institute of Meteorology (INMET; http://www.inmet.gov.br/portal/). The creation of ANA in 2000 led to the release of open hydrological information, which prompted the growth of hydrological studies and fostered water resources management. However, the use of data provided for

Brazilian catchments is challenging because (i) it requires either manual data acquisition of one station at a time through the



institutions' local repositories (e.g., ANA, 2019a) or web scraping techniques to access these data in an automated fashion, (ii) there is little consistency in data format across regions and stations, and (iii) current datasets do not systematically provide catchment attributes characterizing the hydroclimate, landscape, and anthropogenic influences. Further, the difficulty of accessing national meteorological daily time series has led users to compute them from other gridded global databases (e.g.,

Xavier et al., 2016; Beck et al., 2017c; Sun et al., 2018). All these difficulties hinder large-sample hydrological studies in Brazil where, unsurprisingly, nationwide studies (e.g., Siqueira et al., 2018; Bartiko et al., 2019) are less common than in North America or Europe. Consequently, studies in Brazil generally include only a reduced number of stream gauges and catchment attributes, and are restricted to specific regions, such as the Amazon (e.g., Tomasella et al., 2011; Paiva et al., 2013; Latrubesse et al., 2017; Levy et al., 2018), or the La Plata basin (e.g., Collischonn et al., 2001; Pasquini and Depetris, 2007; Melo et al.,

2016; Lima et al., 2017; Chagas and Chaffe, 2018).

To overcome these limitations, we produced and made publicly available a new dataset for large-sample hydrological studies in Brazil, CAMELS-BR. It includes daily streamflow time series from 3,713 stream gauges, daily meteorological time series for 3,097 catchments, and 63 catchment attributes from properties such as topography, climate, land cover, geology, soil, and human intervention. All catchment attributes and time series are in an easily readable file format and on a quickly accessible

database. We follow standards defined by the previous CAMELS and CAMELS-CL datasets, thus allowing direct comparisons with them. Most attributes rely on data products that cover the whole of South America, so they are spatially consistent across Brazil. To reduce the risk of data misinterpretation, we describe the major limitations of the data sources and indices computed. By synthesizing hydrological information from thousands of catchments in Brazil into a single dataset, we allow researchers to skip the arduous task of collecting and preprocessing large quantities of disparate data.

## 2 Hydrometeorological daily time series

### 2.1 Streamflow data

We provide daily streamflow time series for two sets of gauges (Table 1). The first set comprises 3713 streamflow gauges and is provided by the Brazilian National Water Agency (ANA, 2019a). We refer to these as "raw streamflow" time series, as they are readily available from ANA (2019a). Their values are unchanged, but we provide these time series in a different file format

to ease their processing. ANA estimates daily streamflow either by (i) taking two daily stream stage measurements, one in the morning (at 7 am) and another in the afternoon (at 5 pm), which are averaged and transformed into discharge using a stage-discharge relationship (rating curve); or (ii) resorting to regionalization methods when no stream stage measurements are available (no further details on the methods are provided by ANA). The raw streamflow time series cover different periods, ranging from a few days to more than a century. Additionally, although ANA performs data quality checks, these time series

include inconsistencies such as typographical errors and days with missing data. The 3713 gauges are irregularly distributed throughout the country (Fig. 1a). Overall, their spatial distribution is denser and their time series longer in the Southern Atlantic, Southeastern Atlantic, and Paraná hydrographic regions (Fig. 1a and 1b).



The second set of streamflow time series includes 897 gauges, and here we simply refer to them as "streamflow" time series (Table 1). This is the set of gauges used to compute the catchment attributes. It is a subset from the previous 3713 gauges,

which resulted from two selection criteria. Firstly, we selected only gauges that have less than 5 % of missing streamflow data between the water years (starting 1st September) 1990 and 2009. We chose the water years from 1990 to 2009 because (i) it is the period with the largest number of stream gauges with available data (Fig. 2), and (ii) it coincides with the period of analysis from other CAMELS datasets (Addor et al., 2017; Alvarez-Garreton et al., 2018), allowing for direct comparisons with them. Secondly, we only considered catchments for which boundaries have been delimited by Do et al. (2018) and for which there

is a good match with the area estimated by the data provider (see Sect. 3). Although the hydrological signatures introduced below were computed using data from 1990 to 2009, the time series for the 897 stream gauges include data from 1980 to 2018 when available, to enable complementary analyses by other users.

We individually screened the streamflow time series between 1990 and 2009 for errors such as zeroes or repeated values instead of missing values, abrupt changes resulting from changes in measurement instruments or rating curves, annual

streamflow larger than annual precipitation and unrealistic daily streamflow values (i.e., larger than 1,000 mm day$^{-1}$). Gauges affected by such errors were not included in the set of 897 catchments. In addition, we summarized the streamflow metadata provided by ANA as follows. For each daily streamflow measurement, we provide two pieces of information (Table 1). The first metadata variable, "qual_control_by_ana", was set to 1 if the data was quality checked by ANA and to 0 otherwise. The second metadata variable, "qual_flag", indicates the reliability of streamflow estimates. It is also provided by ANA and consists

of the following quality flags: 0, when there is no description; 1, streamflow resulted from stream stage measurements and the rating curve; 2, streamflow estimated by ANA without stream stage measurements; 3, streamflow values marked as doubtful; and 4, when the stream water level falls outside the range of the stream stage, e.g., when the river ran dry. To summarize the metadata (i.e., q_qual_control_frac and q_stream_stage_frac; Table 2), 80 % of the 897 gauges had at least 90 % of their data over 1990-2009 checked for inconsistencies (Fig 1c). The Amazon, São Francisco, and Paraná regions have the lowest

frequency of quality controls in Brazil. Furthermore, the streamflow estimates from 64 % of the 897 catchments were derived from stream stage measurements for 90 % of the days over 1990-2009 (Fig. 1d).

**2.2 Meteorological data**

Meteorological daily time series data are provided for 897 catchments (Table 1). These include (i) precipitation from CHIRPS v2.0 (Funk et al., 2015), CPC (NOAA, 2019a), and MSWEP v2.2 (Beck et al., 2019); (ii) potential evapotranspiration from

GLEAM v3.3a (Miralles et al., 2011; Martens et al., 2017); (iii) actual evapotranspiration from GLEAM v3.3a and MGB South America (Siqueira et al., 2018); and (iv) minimum, maximum, and average temperature from CPC (NOAA, 2019b). The datasets were selected because of their high spatial resolution, their full coverage of South America allowing for consistency through all catchments, and because they are commonly used which enables comparisons with other studies. The daily values represent the average of all cells with their centroids intersected by the catchment (no weight was applied if a cell is only

partially covered by the catchment). A significant limitation of the meteorological data is that, because the cell grids of the



adopted products have resolutions range from 0.05º (ca. 5.5 km² at the equator) to 0.5º (ca. 55 km² at the equator), some catchments are smaller than a single cell. This limitation has to be kept in mind when using the CPC precipitation data (resolution of 0.5º, NOAA, 2019), as precipitation is the meteorological variable with the highest spatial heterogeneity amongst those used in CAMELS-BR.

In addition to GLEAM v3.3a, estimates of actual evapotranspiration were obtained from the MGB model version for South America (Siqueira et al., 2018). The MGB is a conceptual, semi-distributed hydrologic-hydrodynamic model that discretizes the basin (or a set of basins) into irregular unit-catchments and further into hydrological response units by combinations of land use and soil types, where both water and energy balance are computed. The model calculates ET using the Penman-Monteith equation based on CRU meteorological data (i.e., temperature, pressure, radiation, and wind speed) and MSWEP

v1.1 precipitation data (Beck et al., 2017b). Surface resistance is adjusted according to the availability of water in the soil that is updated during the water budget. The MGB also computes the evaporation of flooded areas and intercepted water from the canopy with the Penman equation. Regular ET cells of 0.5º resolution were generated by aggregating unit-catchments using their areas as weights.

## 3 Topographic indices

Even though ANA (2019a) provides estimates of the areas of most gauged Brazilian catchments, the catchment boundaries are not publicly available. Hence, in this study we used the catchment boundaries provided by Do et al. (2018), who used the HydroSHEDS 15 arc-sec resolution digital elevation model (DEM) and delineated the catchments with a procedure similar to Lehner (2012) for more than 3,000 gauges in Brazil. For each streamflow gauge, Do et al. (2018) positioned the outlet at the center of all the DEM grid cells within a radius of 5 km from the gauge coordinates indicated by the metadata. They then

selected the grid cell (and associated catchment boundaries) leading to the catchment area most similar to the one indicated by ANA (2019a). The main limitation of the procedure of Do et al. (2018) is that catchment boundaries were not manually inspected.

Using those catchment boundaries, we computed four topographic attributes (Table 2), namely gauge elevation, catchment mean elevation, mean slope, and area. The area of the catchments ranged from 10.8 km² (i.e., in the upper São Francisco

hydrographic region) to 4.7 million km² (i.e., the Amazon basin at Óbidos). Approximately 30 % of the analyzed catchments are smaller than a thousand km², 43 % are between 1 and 10 thousand km², and 27 % are larger than 10 thousand km². The largest basins are in the Amazon and in the Tocantins-Araguaia hydrographic regions (Fig. 3a). Combined with the Paraguay basin, those regions are usually characterized by low elevations (Fig. 3b), flat slopes (Fig. 3c), and large proportions of wetlands (see Sect. 6.2). The smaller catchments are along the mountain belts on the eastern coast of Brazil, particularly in the

southern and southeastern parts of the country. Those are also the catchments with the steepest slopes. Additionally, many catchments with intermediate elevation ranges (i.e., between 500 and 900 m) are in the central part of the country, which





comprises the Brazilian highlands. Note that, since we computed the average attribute value (unless otherwise noted) of each catchment, the attributes become less representative as the area of the catchment increases.

## 4 Climatic indices

### 4.1 Data and methods

We computed the same eleven climatic indices (Table 3) over the same period (1990 to 2009) as the ones in CAMELS (Addor et al., 2017) and CAMELS-Chile (Alvarez-Garreton et al., 2018). The first water year starts on 1st September 1989 and the last one finishes on 31st August 2009. This is to facilitate inter-dataset comparability. We used precipitation data from CHIRPS v2.0 (Funk et al., 2015) to compute the indices, since it has the highest spatial resolution among the three adopted precipitation

products (i.e., CHIRPS v2.0, CPC, and MSWEP v2.2) and relies on both remote-sensing and gauge-based data.

The mean precipitation, mean potential evapotranspiration, and the aridity index are considered to capture long-term climatic conditions. The aridity index is the ratio of mean potential evapotranspiration to mean precipitation, which stands as a first-order control on the partitioning of precipitation into streamflow (Budyko et al., 1974; Blöschl et al., 2013). Those indices are complemented by the precipitation seasonality index, computed by fitting the seasonal precipitation and temperature using

sine curves representing the seasonal amplitude and time of the year when most of the precipitation occurs (Woods, 2009). The indices of extreme climatic conditions include the frequency, duration, and the most common season of high precipitation events and dry days. Dry days are defined as days with precipitation less than 1 mm, so that the index is not compromised by underdetected precipitation events (Haylock and Nicholls, 2000).

### 4.2 Spatial variability in climatic indices

The mean daily precipitation in Brazil is highest in the Amazon and in Southern Brazil, where it usually exceeds 5 mm day[-1] (1825 mm year[-1]) (Fig. 4a). The lowest mean precipitation occurs in Northeastern Brazil, which is also where mean potential evapotranspiration exceeds the mean precipitation (aridity index > 1, Fig. 4b). The precipitation regime is highly seasonal in most of the country, particularly in the central-west and southeastern Brazil (Fig. 4c). This seasonality is regulated by the South American Monsoon System (Raia and Cavalcanti, 2008; Carvalho et al., 2011), with peaks in the austral summer (Fig. 4f) and

several dry months during the austral winter (Fig. 4i). Southern Brazil has a distinct regime, with a uniform precipitation throughout the year caused by a combination of large-scale phenomena and a diversity of sources of atmospheric moisture (Seager et al., 2010; Martinez and Dominguez, 2014). The Amazon basin, which extends into both hemispheres, has contrasting precipitation regimes between the north (with a peak in austral winter) and the south (with a peak in austral summer) related to alternating warming of each hemisphere (Marengo and Espinoza, 2016). This seasonality is substantially diminished

downstream in the Amazon.

The number of high precipitation and dry days is highest along the catchments on the coast (Fig. 4d and 4g), which is also where the smallest catchments are located. Both indices are significantly correlated with catchment area (p-value < 0.001), so





a regional analysis of both indices should be carried out with caution since large catchments are located in the Amazon and Tocantins-Araguaia basins. On the other hand, the duration of high precipitation (Fig. 4e) and dry day events (Fig. 4h) do not

correlate with catchment area. Their spatial distribution is remarkably similar to the aridity index, except for the Tocantins-Araguaia basin, which has long dry periods but not necessarily long high precipitation events. Summer is the most common season of extreme precipitations in the majority of Brazil, with two main exceptions (Fig. 4f): (i) part of the coast of Northern Brazil; and (ii) Southern Brazil. This is possibly linked to mesoscale convective systems over Southeastern South America (Salio et al., 2007), to sea surface temperature anomalies in the Atlantic ocean (Liebman et al., 2010), and to the El Niño

Southern Oscillation phenomenon, as those regions are particularly affected by it (Grimm, 2011; Tedeschi et al., 2013).

## 5 Hydrological signatures

### 5.1 Data and methods

We computed thirteen hydrological signatures (Table 4) that represent a wide range of hydrological information for the water years from 1990 to 2009. Intermediate streamflow conditions were evaluated with the mean daily flow and its ratio to mean

daily precipitation. These were complemented by baseflow information, a fundamental component that sustains streamflow during dry periods (Smakhtin, 2001). The baseflow index is the ratio of long-term baseflow to long-term total streamflow. We used the digital filter from Ladson et al. (2013) to separate the baseflow component from the hydrograph. The variability of streamflow was evaluated with the slope of the flow duration curve and the streamflow elasticity indices. The slope of the flow duration curve is defined as the slope between the log-transformed 33rd and 66th long-term percentiles of daily streamflow

(Yadav et al., 2007; Sawicz et al., 2011). High values of that index suggest highly variable streamflow, caused either by a high seasonality of streamflow or by a flashy response to precipitation events (Yokoo and Sivapalan, 2011; McMillan et al., 2017). Streamflow elasticity is an indicator of the sensitivity of mean annual flow to changes in mean annual precipitation (Sankarasubramanian et al., 2001). For example, a streamflow elasticity value of 2 indicates that a 1 % change in mean annual precipitation generates a 2 % change in mean annual flow. Extreme streamflow conditions were analyzed using signatures

based on the magnitude, frequency, and duration of high and low flow events. High and low flow events were defined through long-term thresholds, based on the median and mean flow, respectively (Olden and Poff, 2003). The magnitude of high and low flow events was characterized using the 5th and the 95th percentiles. There are two primary limitations to the hydrological signatures used in this study. First, several signatures might scale with catchment area. Since catchment area varies substantially among hydrographic regions, spatial analyses should be carefully conducted. Second, we did not check for

temporal dependencies of consecutive high or low flow events, for example when two flood peaks occur within a couple of days from each other and both may be related to a single extreme precipitation event. Many criteria exist to identify independent high flow events (Hall et al., 2014; Archfield et al., 2016) and low flow events (Fleig et al., 2006; Van Loon, 2015), which might lead to differences in the analyzed signatures.



## 5.2 Spatial variability in hydrological signatures

The spatial distribution of mean daily flows (Fig. 5a) and runoff ratio (Fig. 5b) closely resembles that of mean daily precipitation. These are notably high in Southern Brazil and in parts of the Amazon; and low in Northeastern Brazil. The mean half-flow date follows a gradient ranging from February and March in the Eastern Atlantic region to May in the Amazon and on the northern coast (Fig. 5c). Steep slopes of the flow duration curve occur especially in the tributaries of Southern Amazon, the Tocantins-Araguaia basin, the Eastern Atlantic hydrographic region and in parts of Southern Brazil (Fig. 5d). Some

catchments have undefined values, meaning that they have zero flow for more than 33 % of the time. Since the slopes of the flow duration curve indicate the overall streamflow variability, they are spatially similar to several other hydrological signatures. They are, most noticeably: (i) negatively correlated with the baseflow index (Fig. 5e), hence catchments with high baseflow may be highly resilient to dry periods (Fan, 2015); (ii) positively correlated with streamflow precipitation elasticity (Fig. 5f), which indicates variability at the interannual timescale; (iii) negatively correlated with the 5th percentile of

streamflow (i.e., low flows; Fig. 5l); and (iv) positively correlated with the frequency and duration of low flow events (Fig. 5j and 5k). However, note that some regions do not follow those patterns. In particular, catchments in Southern Amazon and in the Tocantins-Araguaia basin have high baseflow indices despite steep slopes of the flow duration curve. It possibly implies that the variability in those catchments is related to a high seasonality, rather than to a flashy response to precipitation events. High flow days are more frequent and their events are longer in Southern Brazil, in the Eastern Atlantic region, and on the

coast of northeastern Brazil (Fig. 5g and 5h). Those regions also have the most frequent and longest low flow events. This suggests that, in addition to the catchments in Southwestern Amazon and in the Tocantins-Araguaia basin, the extremes of both high and low flows might be related. Catchments seldom have high values in all three high flow signatures, except for Southern Brazil, revealing that this might be the region with the most problematic flood episodes. On the other hand, long and frequent low flows are found in nearly all hydrographic regions. The majority of catchments in the Eastern Atlantic and the

East Northeastern Atlantic regions are characterized by long and frequent low flows, where nearly half of those have at least 100 days of low flows in the year.

## 6 Land cover characteristics

### 6.1 Data and methods

Each catchment was described using ten land cover classes (Table 5) based on GlobCover2009 (Arino et al., 2012).

GlobCover2009 uses imagery from Envisat's Medium Resolution Imaging Spectrometer Instrument Fine Resolution (MERIS FR). The classification has a spatial resolution of 300 m, a global coverage every three days, and is based on images from January until December 2009. GlobCover2009 classification includes 22 land cover classes but, to simplify the dataset, we combined these into similar classes. In particular, the class "forests" is a combination of broadleaved and needleleaved forests, either evergreen or deciduous. Note that GlobCover 2009 does not differentiate between natural or planted forests.





There are three main limitations of the GlobCover2009 dataset. High confusion rates between croplands and grasslands show that the separation of crops, pastures, and meadows can be problematic, especially in Brazil where those land covers occur extensively. Identification of wetlands is also an issue (Arino et al., 2012) and flooded forests might be underrepresented in the classification. Lastly, GlobCover2009 used the Shuttle Radar Topography Mission (SRTM) Water Body Data, which is based on data from 2000 and does not coincide with the 2009 MERIS data.

## 6.2 Spatial variability in land cover characteristics

Croplands are widespread in Brazil, especially in the highlands, in Southern Brazil, and on the eastern coast of Northeastern Brazil (Fig. 6a and 6d). Out of the 897 CAMELS-BR catchments, 52.4 % have croplands or mosaics of croplands and natural vegetation as the dominating landcover (Fig. 6c). Croplands are most noticeable particularly in the Uruguay and Paraná hydrographic regions. Even though GlobCover2009 does not cover the same period as the hydrological signatures (i.e., 1990-

2009), croplands were already extensive in almost all states in Brazil in the 1980s and pastures in the 1960s (Leite et al., 2012; Dias et al., 2016). This is true except for Southern Amazon, where agricultural expansion has led to one of the highest deforestation rates in the world since the 1980s (Song et al., 2018).

Aside from the Amazon, catchments dominated by forests are located in mountain belts, i.e., in steep slope regions in Southern and Southeastern Brazil (Fig. 6b). Shrublands occur mainly in the driest regions of the country (Fig. 6e), but they are not the

predominant land cover in these regions. Natural wetlands or water bodies are largely present in the Amazon, Tocantins-Araguaia, and Paraguay hydrographic regions (not shown). Some catchments in the Paraná, Uruguay, and São Francisco basins are also substantially covered by water bodies. However, those are mainly artificial reservoirs (see Sect. 9.3). The CAMELS-BR catchments typically have a low fraction of their area considered to be "impervious areas", such as urban land covers; only 0.2 % of the catchments have more than 5 % of impervious areas (not shown). In addition, grasslands, bare soil areas, and

permanent snow are rare in the CAMELS-BR catchments (not shown).

## 7 Geologic characteristics

### 7.1 Data and methods

The geology of the catchments was described using seven geologic attributes (Table 6). The first and second most common geologic class, their fractions, and the percentage of the catchment covered by carbonate rocks were extracted from the Global

Lithological Map (GLiM; Hartmann and Moosdorf, 2012). GLiM was created by assembling information from 92 regional lithological maps. In the Brazilian territory, it relies on data from the Brazilian Geological Survey at the 1:1 million scale (Schobbenhaus et al., 2004). We considered only the first level of the GLiM geologic classes, which classifies lithology into 16 groups. The additional second and third levels provide more specific geologic information but were not included in this study. We note that two geologic classes cover a particularly broad variety of rocks. First, the "unconsolidated sediments"

class is quite unspecific with regards to the sediment types and grain sizes (it includes sediments originated by areas as alluvial,





swamp, and dune deposits). Second, catchments dominated by the "metamorphic rocks" class can have a wide range of lithologies, from shales to gneiss and quartzite.

We extracted the subsurface permeability and porosity indices from the GLobal HYdrogeology MaPS 2.0 (GLHYMPS; Gleeson et al., 2014; Huscroft et al., 2018), which is modeled based on information from the GLiM and the Global

Unconsolidated Sediments Map (GUM; Börker et al., 2018). Subsurface permeability indicates how easily water can flow through the subsurface. GLHYMPS modeled it only for saturated conditions (Huscroft et al., 2018), so it is not adequate to characterize regions dominated by unsaturated processes, e.g., deeply weathered soils. The subsurface porosity indicates the fraction of void spaces in a material and controls the water storage capacity in the subsurface. A major caveat of GLHYMPS data is that it is only adequate for analyses at the regional scale, i.e., over spatial units greater than 5 km (Gleeson et al., 2014).

**7.2 Spatial variability in geological characteristics**

The catchments on the eastern coast have lithologies dominated by either metamorphic or acid plutonic rocks (Fig. 7a and 7b), related to high elevation and steep slopes in this region. These catchments also have low subsurface permeability (Fig. 7g) and the lowest subsurface porosity rates in the country (Fig. 7f). In Southern Brazil, basic volcanic lithology is widespread, which encompasses basaltic rocks. Southern Brazil has the most homogeneous lithological types of the country (Fig. 7c and 7d),

where more than 80 % of the catchment areas are usually characterized by a single lithological type. However, subsuperficial porosity and permeability are highly heterogeneous, extending from middle-range to high porosity values and from middle-range to low permeability values.

Sedimentary rocks occur on a large scale at São Francisco, Parnaíba, Western Northeast Atlantic, and part of the Amazon hydrographic regions. The Northern Amazon is characterized mostly by metamorphic or plutonic rocks, while the Western

Amazon has either siliciclastic or mixed sedimentary lithologies. On the other hand, unconsolidated sedimentary lithologies occur particularly downstream in the Amazon, the Tocantins and the Paraguay basins. These basins also have flat slopes and large proportions of wetlands, which allows for alluvial particles to settle down. Most of the catchments with a high proportion of sedimentary rocks have high subsurface porosity, although their permeability varies according to the grain sizes of these rocks. Carbonate sedimentary rocks, such as karst or limestone, are more common in the São Francisco basin and in western

Amazon (Fig. 7e). Those rocks are also present in some isolated and smaller catchments in Southern Brazil, in the Paraguay, and in the Tocantins-Araguaia basins.

**8 Soil characteristics**

**8.1 Data and methods**

We provide six soil characteristics (Table 7). Five of those were extracted from SoilGrids250m (Hengl et al., 2017; Shangguan

et al., 2017), a collection of soil maps for the world at the 250 m resolution. SoilGrids250m maps are the result of a model using approximately 150,000 soil profiles, with predictions based on machine learning methods and 158 remote-sensing



covariates including climate, vegetation, geomorphology, and lithology (Hengl et al., 2017). Although SoilGrids250m generated predictions for several soil depths, in this work we only computed soil characteristics over a depth of 30 cm. SoilGrids250m is based on a machine-learning model that explains large proportions of the variance of most observed variables, including 69 % of the variance of organic carbon content and more than 70 % of the soil textures (i.e., clay, silt, and sand content).

The soil characteristics might be highly correlated with other attributes from CAMELS-BR since they are modeled based on climatic and landscape covariates. Organic carbon content and clay content have modeled depth to bedrock as a predominant variable (Hengl et al., 2017). Other variables are also important, such as temperature and geomorphological characteristics (e.g., surface slope). The predictions of sand content are based primarily on depth to bedrock and precipitation, both at similar weights. Out of the five variables considered from the SoilGrids250m, predictions of depth to bedrock is the most problematic, with 59 % of its variance explained by the model (Shangguan et al., 2017). It has precipitation as the predominant covariate, which accounts for the control of weathering rates and soil production. Other decisive covariates are vegetation dynamics and geomorphological characteristics, which accounts for factors such as soil erosion (Shangguan et al., 2017).

The sixth soil characteristic is the water table depth, based on a 1 km resolution global model by Fan et al. (2013). Combined with depth to bedrock, water table depth can be an indicator of water storage potential in the catchment, which is related to baseflow and the supply of water for the vegetation during dry periods (Fan et al., 2013, 2019). The most important variables in the predictions of the water table depth of that model are, in decreasing order of importance, surface slope, elevation, precipitation and temperature (Fan et al., 2013). Note that groundwater abstractions are not represented in the model, so water table depth data must be used with caution when analyzing catchments with intense anthropogenic intervention.

## 8.2 Spatial variability in soil characteristics

Soil texture in CAMELS-BR is characterized by (i) a predominance of clay content in Southern Brazil, in parts of Southeastern Brazil, particularly in higher elevations, and in northeastern Amazon (Fig. 8c); (ii) similar values of clay, sand, and silt content in the southern tributaries of Western Amazon (Fig. 8a to 8c); and (iii) a wide predominance of sand content in the rest of the country (Fig. 8a). As expected, the aridity index is closely related with the spatial distribution of the soil texture, since climatic attributes are important covariates in SoilGrids250m predictions (Hengl et al., 2017). The predominance of clay in Southern Brazil and in part of Southeastern Brazil might be linked to their lithological classes, i.e., with basic volcanic rocks in the former and acid plutonic rocks in the latter, since they have coincidental spatial distributions.

Organic carbon content is most pronounced in parts of the Amazon and in regions with high clay content (Fig. 8d). The depth to bedrock is higher in Central and Northeastern Brazil, frequently above 30 m (Fig. 8e). On the other hand, only 14 % of the catchments have depths to bedrock lower than 10 m, all of them located in Southern Brazil. Regarding water table depths, there is a clear gradient with higher depths on the eastern coast of Brazil (i.e., exceeding 30 m deep) to lower depths towards the Amazon (i.e., less than 10 m deep). Amongst all six soil characteristics indices, water table depth has the lowest



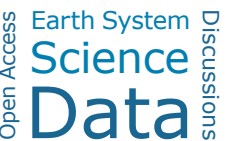

correspondence to climate, i.e., to mean precipitation and aridity index. It is mostly correlated with catchment slopes, as
previously indicated by Fan et al. (2013).

## 9 Human intervention indices

### 9.1 Data and methods for consumptive water use

We computed four indices of human intervention in the catchments (Table 8). Two are the total consumptive water use in the
catchment in 2017, one normalized by catchment areas and another normalized by mean annual streamflow. The water uses
are based on the Manual of Consumptive Water Use in Brazil (ANA, 2019c), which estimated the monthly water use of each
municipality in Brazil. These estimates are the sum of water demands from six categories:

(i) Irrigation: demand based on water balance models that estimate the quantity of water needed by irrigated crops but not
supplied by precipitation or soil moisture ANA (2019c). The spatial extent of irrigated croplands was characterized using the
national censuses of agriculture (e.g. IBGE, 2007) and remote-sensing images from Landsat, Sentinel-2, and Moderate-
Resolution Imaging Spectroradiometer (MODIS) (ANA, 2019b).

(ii) Livestock: demand estimated by multiplying the number of livestock units with their corresponding daily drinking
requirements. The number and type of livestock of each municipality were mapped on the national censuses of agriculture.

(iii) Households: demand estimated by multiplying the number of people in a municipality by their per capita domestic water
use.

(iv) Industry: demand estimated by multiplying the number of employees in several industrial categories from each
municipality by its per capita water use.

(v) Mining: demand estimated by combining the water use coefficient with the annual production of several types of mineral
extraction.

(vi) Thermoelectricity: demand estimated by applying a water use coefficient to the annual electricity production of each
thermoelectric plant in the country.

These water demand estimates do not differentiate surface water from groundwater. Even though groundwater abstraction is
extensive in the eastern part of Brazil (Fan et al., 2013), it is estimated that most of the water use in South America comes
from surface water (Wada et al., 2014). To estimate the consumptive water use of each catchment, we divided the values of
each municipality by its area. We assumed the water use to be spatially homogeneous throughout the municipality territory
and transferred the data for each municipality onto a 500 m spatial resolution raster.

There are three major limitations of using ANA (2019c) estimated consumptive water use. First, evaporation from artificial
reservoirs was not included in the computation. Thus, water use might be underestimated, particularly in the northeastern part
of Brazil, i.e., in the driest part of the country. Second, the dataset comes on an irregular grid, since municipalities areas vary





significantly. The smallest municipalities are usually within 500 km of the coast and their areas are mostly a few hundred km². In contrast, the western part of Brazil and the Amazon usually have municipalities larger than a thousand km². Hence, consumptive water use of small catchments in the western part of Brazil should be interpreted with caution because they are smaller than the input data. The third limitation is that the consumptive water use of South America outside Brazil was not

estimated by ANA (2019c) and was not considered in this study. This affects particularly the basins in the Amazon since they cover large parts out of Brazil. That said, anthropic intervention in these basins is low: only three basins with international borders in the Amazon are more than 10 % covered by croplands or croplands and natural vegetation mosaic; and none has more than 0.05 % of impervious land covers such as urban areas.

### 9.2 Data and methods for reservoirs

The other two indices for human intervention are related to flow regulation (Table 8), i.e., the sum of the total storage capacity of all reservoirs in the catchment and its ratio to the total annual flow of the catchment (i.e., the degree of regulation). We worked with estimated storage capacities from 1406 reservoirs in South America. The reservoirs were mapped by combining three data sources: (i) the Global Reservoirs and Dam database v1.3 (GRanD; Lehner et al., 2011); (ii) the hydroelectric power plants database of the National Electrical System Operator (ONS, 2019); and (iii) the 2017 National Dam Safety Report (ANA,

2018) database. The GRanD database includes reservoirs throughout South America, while the other two provided data only for Brazil. The procedure for combining the three databases was:

(i) We included all reservoirs from GRanD v1.3 in South America.

(ii) For each GRanD reservoir, we visually compared the inundated area with the one indicated by the polygons from the water

bodies maps from Pekel et al. (2016). When the inundated areas differed substantially, we substituted the former with the latter and updated the size of the inundated area.

(iii) Out of more than 24,000 reservoirs from ONS (2019) and ANA (2018) databases, we included only those that have their inundated areas (Pekel et al., 2016) visible at the 1:500,000 scale. Although our goal was to only include reservoirs larger than approximately 0.5 km², some smaller reservoirs were also included. We computed the size of the inundated areas of those

reservoirs according to the polygons from Pekel et al. (2016).

(iv) To check for duplicates in the databases, we manually inspected all dam points and their inundated areas.

(v) Finally, the storage capacities of reservoirs updated in step (ii) or included in step (iii) were recalculated using their inundated areas and, when available, information on dam height. We applied two equations determined by Lehner et al. (2011, Technical Documental) with a statistical regression using data from 5824 reservoirs worldwide. When information on dam

height was available, we applied Eq. (1):

$$V = 0.678 \, (A \, h)^{0.9229} \, , \tag{1}$$





where V is the reservoir storage capacity in $10^6$ m$^3$; A is the size of the inundated area in km$^2$; and h is dam height in m. When information on dam height was not available, we used Eq. (2):

$$V = 30.684 \, A^{0.9578} \tag{2}$$

### 9.3 Spatial variability in human intervention indices

The spatial distribution of human interference indices reveals that, unlike the catchments in the original CAMELS for the United States, catchments in CAMELS-BR can be significantly impacted by human activities. There are 17.8 % of catchments with annual consumptive water uses greater than 5 % of the mean annual flow. Those are principally in the driest parts of the country, i.e., in the São Francisco, Eastern Atlantic, Eastern Northeast Atlantic, and upper Paraná hydrographic regions (Fig. 9b). Nevertheless, water uses greater than 20 % of the mean annual flow are rare, occurring in only 3.9 % of the catchments. The similarity encountered between arid climates and high consumptive water uses may be attributed to two main causes. First, in the most arid catchments, the mean annual flow is typically a third of that of the rest of the country, which, unsurprisingly, leads to higher water uses proportional to the annual flow. Second, crops in drier climates require frequent irrigation and considerable rates of water withdrawal. On the other hand, we observe that the central and southeastern regions of Brazil have the greatest values of water uses normalized by catchment area (Fig. 9a). Catchments in those regions are commonly occupied by either irrigated croplands or populous metropolitan areas, which are respectively the first and second categories with the highest water demands in Brazil (ANA, 2019c).

The degree of regulation is related to catchment area (Fig. 9c), meaning that the most regulated basins are downstream in the river basins. The main rivers with high regulations are the Paraná, Uruguay, São Francisco, Tocantins-Araguaia, Parnaíba, and Paraíba do Sul rivers. In those regions,19.2 % and 7.2 % of the catchments have a degree of regulation greater than 10 % and 50 %, respectively. These values nearly double in the driest regions of the country (i.e., the Eastern Atlantic, São Francisco, Eastern Northeast Atlantic, and Parnaíba hydrographic regions): 37.6 % and 22.1 % of the catchments have a degree of regulation greater than 10 % and 50 %, respectively. Therefore, the driest catchments of CAMELS-BR dataset have the highest human intervention rates, both in terms of consumptive water use and reservoir regulation

### 10 Data availability

The CAMELS-BR dataset is freely available at https://doi.org/10.5281/zenodo.3709338 (Chagas et al., 2020). The files provided are (i) the 63 attributes in a zip file, (ii) the daily time series in zip files, (iii) the catchment boundaries used to compute the attributes and extract the time series, computed by Do et al. (2018) and Gudmundsson et al. (2018), and (iv) a readme file.

## 11 Conclusions

So far, large-sample hydrological studies in Brazil lacked a comprehensive and easily accessible dataset. Here, we introduced the CAMELS-BR, a new dataset comprising more than 3,000 catchments in Brazil. The data provided include daily streamflow and meteorological time series and 63 catchment attributes. The attributes cover a wide range of fundamental properties for large-sample hydrological research, such as topography, land cover, geology, soil, and human intervention characteristics. We strived to make CAMELS-BR as comparable as possible to the other CAMELS datasets (Addor et al., 2017; Alvarez-Garreton

et al., 2018) by using common naming conventions, scripts and datasets. We also discuss the major limitations of the data to limit the risk of misinterpretation and misuse.

Even though CAMELS-BR is a step forward for hydrological research in Brazil, there are several opportunities for expanding the dataset in the future. For example, future versions of CAMELS-BR could include additional catchment attributes critical to understand hydrological processes, such as drainage density and basin morphometry (Shen et al., 2017). Further, an updated

version should better characterize heterogeneities within each catchment, both for the time series and attributes. Additionally, since data uncertainties are omnipresent (Montanari, 2007; Blöschl et al., 2019b; Addor et al., 2019), they should be further explored by including additional data sources.

By simplifying the access to hydrological data, we aim to encourage further large-sample hydrological studies in Brazil, to facilitate the inclusion of Brazilian catchments in global large-sample studies, and to increase the transparency and

reproducibility of these studies. We believe the data introduced here will in particular prove useful to explore the drivers of catchment behavior, to anticipate hydrological changes, and to study the impacts of human activities on the water cycle. We see CAMELS-BR as a resource designed to serve the broad water science community and to help with water resources management at regional, national, and continental scales.

**Author contribution**

PC and VC initiated the investigation. VC, PC, NA, FF, AF, RP, and VS designed the study. VC processed the data and created the figures. VC and NA computed the catchment attributes. VC prepared the manuscript with contributions from all co-authors.

**Competing interests**

The authors declare that they have no conflict of interest.

**Acknowledgements**

VC would like to thank CNPq (Brazilian National Council for Scientific and Technological Development) for the scholarship. NA acknowledges support from the Swiss National Sciences Foundation (P400P2_180791).



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





**Table 1. Summary of the data provided by CAMELS-BR.**

| | Variable | Description | N. of catchments |
|---|---|---|---|
| **Daily time series** | Raw streamflow | As read from the Brazilian National Water Agency[a], in $m^3 s^{-1}$, with varying coverage periods, with data quality flags (qual_control_by_ana and qual_flag) for each time step | 3713 |
| | Streamflow | Transformed to mm day$^{-1}$, covering from 1980-2018, with data quality flags (qual_control_by_ana and qual_flag) for each time step | 897 |
| | Precipitation | Catchment averages (mm day$^{-1}$), with varying coverage periods between 1979-2018, using three products: CHIRPS v2.0 (0.05º res.)[b]; CPC Global Unified (0.5º res.)[c]; and MSWEP v2.2 (0.1º res)[d]. | 897 |
| | Potential evapotranspiration | Catchment averages (mm day$^{-1}$), from 1980-2018, using GLEAM v3.3a (0.25º res.)[e] | 897 |
| | Actual evapotranspiration | Catchment averages (mm day$^{-1}$), with varying coverage periods between 1980-2018, using two products: GLEAM v3.3a (0.25º res.)[e]; and MGB (0.5º res.)[f] | 897 |
| | Minimum, maximum, and average temperature | Catchment averages (degrees Celsius day$^{-1}$), from 1979-2018, using CPC Global Unified (0.5º res.)[g] | 897 |
| **Catchment attributes** | Location | 5 attributes (Table 2) | 3713 |
| | Topography | 4 attributes (Table 2) | 897 |
| | Data quality checks | 2 attributes (Table 2) | 897 |
| | Climatic indices | 11 attributes (Table 3) | 897 |
| | Hydrological signatures | 13 attributes (Table 4) | 897 |
| | Land cover characteristics | 11 attributes (Table 5) | 897 |
| | Geologic characteristics | 7 attributes (Table 6) | 897 |
| | Soil characteristics | 6 attributes (Table 7) | 897 |
| | Human intervention indices | 4 attributes (Table 8) | 897 |

[a] ANA (2019a).

[b] Climate Hazards Group InfraRed Precipitation with Station data v2.0 (Funk et al., 2015).

[c] Climate Prediction Center Global Unified Gauge-Based Analysis of Daily Precipitation (NOAA, 2019a).

[d] Multi-Source Weighted-Ensemble Precipitation (Beck et al., 2019).

[e] Global Land Evaporation Amsterdam Model v3.3a (Miralles et al., 2011; Martens et al., 2017).

[f] Large-Scale Hydrological Model (Siqueira et al., 2018).

[g] Climate Prediction Center Global Daily Temperature (NOAA, 2019b).



**Table 2. Location, topographic characteristics, and data quality checks.**

| Attribute | Description | Units | Data source |
|---|---|---|---|
| gauge_id | Catchment identifier provided by ANA | - | ANA (2019a) |
| gauge_name | Gauge name provided by ANA | - | ANA (2019a) |
| gauge_region | Hydrographic region | - | ANA (2019a) |
| gauge_lat | Gauge latitude | ºN | ANA (2019a) |
| gauge_lon | Gauge longitude | ºE | ANA (2019a) |
| elev_gauge | Gauge elevation | m.a.s.l. | HydroSHEDS 15 arc-sec DEM |
| elev_mean | Catchment mean elevation | m.a.s.l. | HydroSHEDS 15 arc-sec DEM |
| slope_mean | Catchment mean slope | m km$^{-1}$ | HydroSHEDS 15 arc-sec DEM |
| area | Catchment area | km² | Do et al. (2018) |
| q_quality_control_frac | Fraction of streamflow data (1990-2009) with quality control checks by ANA | % | ANA (2019a) |
| q_stream_stage_frac | Fraction of streamflow data (1990-2009) derived from stream stage measurements | % | ANA (2019a) |



**Table 3. Climatic indices.**

| Attribute | Description | Units | Data source |
|---|---|---|---|
| p_mean | Mean daily precipitation | mm day$^{-1}$ | CHIRPS v2.0 |
| pet_mean | Mean daily potential evapotranspiration (PET) | mm day$^{-1}$ | GLEAM v3.3a |
| aridity | Aridity, computed as the ratio of mean PET to mean precipitation | - | GLEAM v3.3a and CHIRPS v2.0 |
| p_seasonality | Seasonality and timing of precipitation, estimated using sine curves to represent the annual temperature and precipitation cycles; positive (negative) values indicate that precipitation peaks in summer (winter); values close to 0 indicate uniform precipitation throughout the year | - | CHIRPS v2.0 |
| frac_snow | Fraction of precipitation falling as snow (i.e., on days colder than 0 ℃) | - | CHIRPS v2.0 and CPC |
| high_prec_freq | Frequency of high precipitation days ($\geq 5$ times the mean daily precipitation) | days yr$^{-1}$ | CHIRPS v2.0 |
| high_prec_dur | Average duration of high precipitation events (number of consecutive days $\geq 5$ times the mean daily precipitation) | days | CHIRPS v2.0 |
| high_prec_timing | Season during which most high precipitation days ($\geq 5$ times the mean daily precipitation) occur | season | CHIRPS v2.0 |
| low_prec_freq | Frequency of dry days ($< 1$ mm day$^{-1}$) | days yr$^{-1}$ | CHIRPS v2.0 |
| low_prec_dur | Average duration of dry periods (number of consecutive days $< 1$ mm day$^{-1}$) | days | CHIRPS v2.0 |
| low_prec_timing | Season during which most dry days ($< 1$ mm day$^{-1}$) occur | season | CHIRPS v2.0 |




**Table 4. Hydrological signatures.**

| Attribute | Description | Units | Data source |
|---|---|---|---|
| q_mean | Mean daily discharge | mm day$^{-1}$ | ANA (2019a) |
| runoff_ratio | Runoff ratio, computed as the ratio of mean daily discharge to mean daily precipitation | - | ANA (2019a) |
| stream_elas | Streamflow precipitation elasticity (i.e., the sensitivity of streamflow to changes in precipitation at the annual timescale, using the mean daily discharge as reference) | - | ANA (2019a) |
| slope_fdc | Slope of the flow duration curve between the log-transformed 33rd and 66th streamflow percentiles | - | ANA (2019a) |
| baseflow_index | Baseflow index, computed as the ratio of mean daily baseflow to mean daily discharge, with the hydrograph separation performed using the Ladson et al. (2013) digital filter | - | ANA (2019a) |
| hfd_mean | Mean half-flow date (i.e., the date on which the cumulative discharge since 1$^{st}$ September reaches half of the annual discharge) | day of the year | ANA (2019a) |
| Q5 | 5% flow quantile (low flow) | mm day$^{-1}$ | ANA (2019a) |
| Q95 | 95% flow quantile (high flow) | mm day$^{-1}$ | ANA (2019a) |
| high_q_freq | Frequency of high-flow days (> 9 times the median daily flow) | days yr$^{-1}$ | ANA (2019a) |
| high_q_dur | Average duration of high-flow events (number of consecutive days > 9 times the median daily flow) | days | ANA (2019a) |
| low_q_freq | Frequency of low-flow days (< 0.2 times the mean daily flow) | days yr$^{-1}$ | ANA (2019a) |
| low_q_dur | Average duration of low-flow events (number of consecutive days < 0.2 times the mean daily flow) | days | ANA (2019a) |
| zero_q_freq | Percentage of days with zero discharge | % | ANA (2019a) |



**Table 5. Land cover characteristics.**

| Attribute | Description | Units | Data source |
|---|---|---|---|
| crop_frac | Percentage covered by croplands | % | ESA GlobCover2009 |
| crop_mosaic_frac | Percentage covered by a mosaic of croplands and natural vegetation | % | ESA GlobCover2009 |
| forest_frac | Percentage covered by broadleaved or needleleaved forests, either evergreen or deciduous | % | ESA GlobCover2009 |
| shrub_frac | Percentage covered by shrublands | % | ESA GlobCover2009 |
| grass_frac | Percentage covered by grasslands or areas with sparse (<15%) vegetation | % | ESA GlobCover2009 |
| bare_frac | Percentage covered by bare areas | % | ESA GlobCover2009 |
| imperv_frac | Percentage covered by artificial surfaces or urban areas | % | ESA GlobCover2009 |
| wet_frac | Percentage covered by water bodies or wetlands | % | ESA GlobCover2009 |
| snow_frac | Percentage covered by permanent snow or ice | % | ESA GlobCover2009 |
| dom_land_cover | Dominant land cover | - | ESA GlobCover2009 |
| dom_land_cover_frac | Percentage covered by the dominant land cover | % | ESA GlobCover2009 |




**Table 6. Geologic characteristics.**

| Attribute | Description | Units | Data source |
|---|---|---|---|
| geol_class_1st | Most common geologic class in the catchment | - | GLiM |
| geol_class_1st_frac | Fraction of the catchment covered by the most common geologic class | - | GLiM |
| geol_class_2nd | Second most common geologic class in the catchment | - | GLiM |
| geol_class_2nd_frac | Fraction of the catchment covered by the second most common geologic class | - | GLiM |
| carb_rocks_frac | Fraction of the catchment covered by carbonate sedimentary rocks | - | GLiM |
| geol_porosity | Subsurface porosity of the catchment | - | GLHYMPS v2.0 |
| geol_permeability | Subsurface permeability (log10 scale) of the catchment, extract for each catchment using the geometric mean | $m^2$ | GLHYMPS v2.0 |





**Table 7. Soil characteristics.**

| Attribute | Description | Units | Data source |
|---|---|---|---|
| sand_frac | Fraction of sand content of the soil material smaller than 2 mm at a depth of 30 cm | % | SoilGrids250m |
| silt_frac | Fraction of silt content of the soil material smaller than 2 mm at a depth of 30 cm | % | SoilGrids250m |
| clay_frac | Fraction of clay content of the soil material smaller than 2 mm at a depth of 30 cm | % | SoilGrids250m |
| org_carbon_content | Soil organic carbon content at a soil depth of 30 cm | g kg$^{-1}$ | SoilGrids250m |
| depth_bedrock | Depth to bedrock | cm | SoilGrids250m |
| water_table_depth | Median water table depth | cm | Fan et al. (2013) |





**Table 8. Human intervention indices.**

| Attribute | Description | Units | Data source |
|---|---|---|---|
| consumptive_use_mm | Total consumptive water use in 2017, normalized by catchment area | mm yr$^{-1}$ | ANA (2019c) |
| consumptive_use | Total consumptive water use in 2017, normalized by mean annual streamflow | % | ANA (2019c) |
| vol_reservois | Total maximum storage capacity of the reservoirs in the catchment | $10^6$ m$^3$ | GRanD v1.3, ONS, and ANA (2018) |
| degree_of_regulation | Ratio of total reservoir storage capacity of the catchment to its total annual flow | % | GRanD v1.3, ONS, and ANA (2018) |

**Figure 1. (a) South America and the total river discharge data availability from the stream gauges included in this study. The black line surrounded by a white line indicates rivers. The dashed line is Brazil's borders. (b) Hydrographic regions of Brazil according to ANA (2019a). (c) Fraction of streamflow data with quality control checks by ANA. (d) Fraction of streamflow data derived from stream stage measurements. The circles are located at the outlet of the catchments and their sizes are proportional to the sizes of the catchments. The grey line in (c) and (d) indicates the limits of hydrographic regions.**


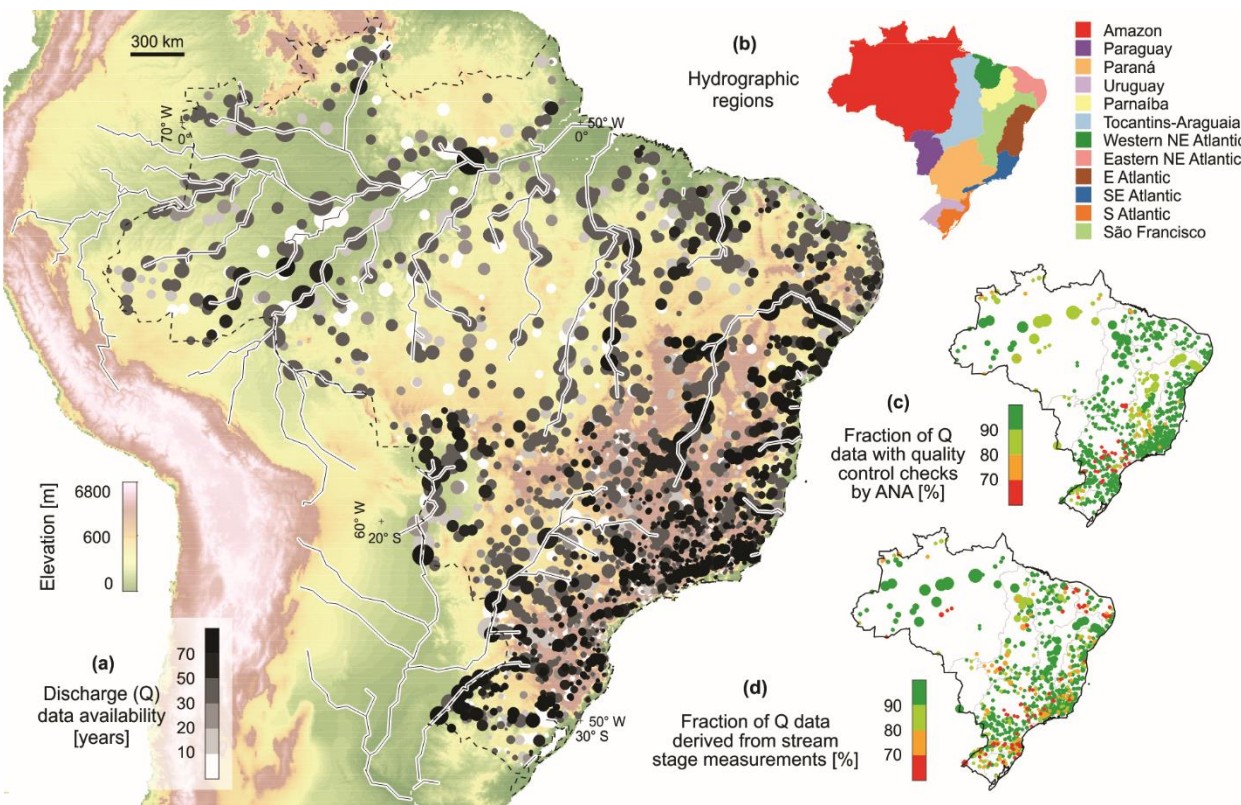



**Figure 2.** Time series with the number of streamflow gauges with at least one measurement for a given year in Brazil.

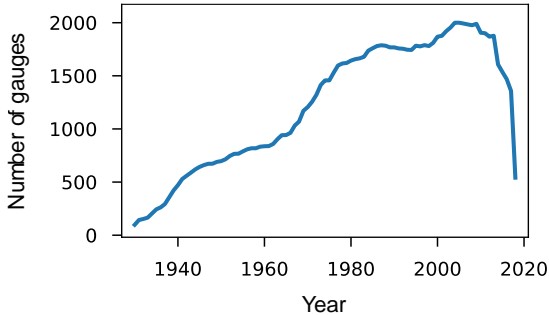




**Figure 3. Topographic characteristics. The size of the circles is proportional to the size of the catchment. The grey line indicates the limits of hydrographic regions.**

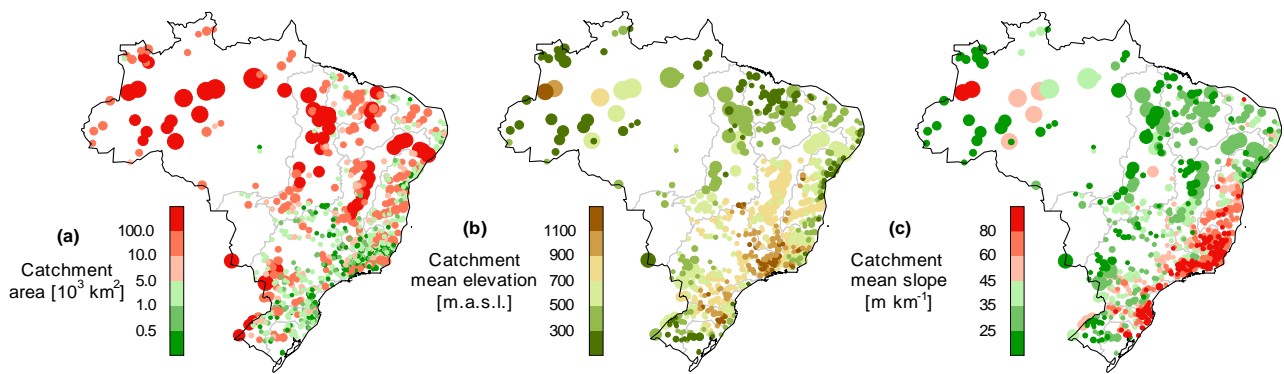



**Figure 4. Climatic indices. The size of the circles is proportional to the size of the catchment. The grey line indicates the limits of hydrographic regions.**

**Figure 5. Hydrological signatures. The black circles are catchments with undefined values. The size of the circles is proportional to the size of the catchment. The grey line indicates the limits of hydrographic regions.**





**Figure 6. Land cover characteristics. The size of the circles is proportional to the size of the catchment. The grey line indicates the limits of hydrographic regions.**

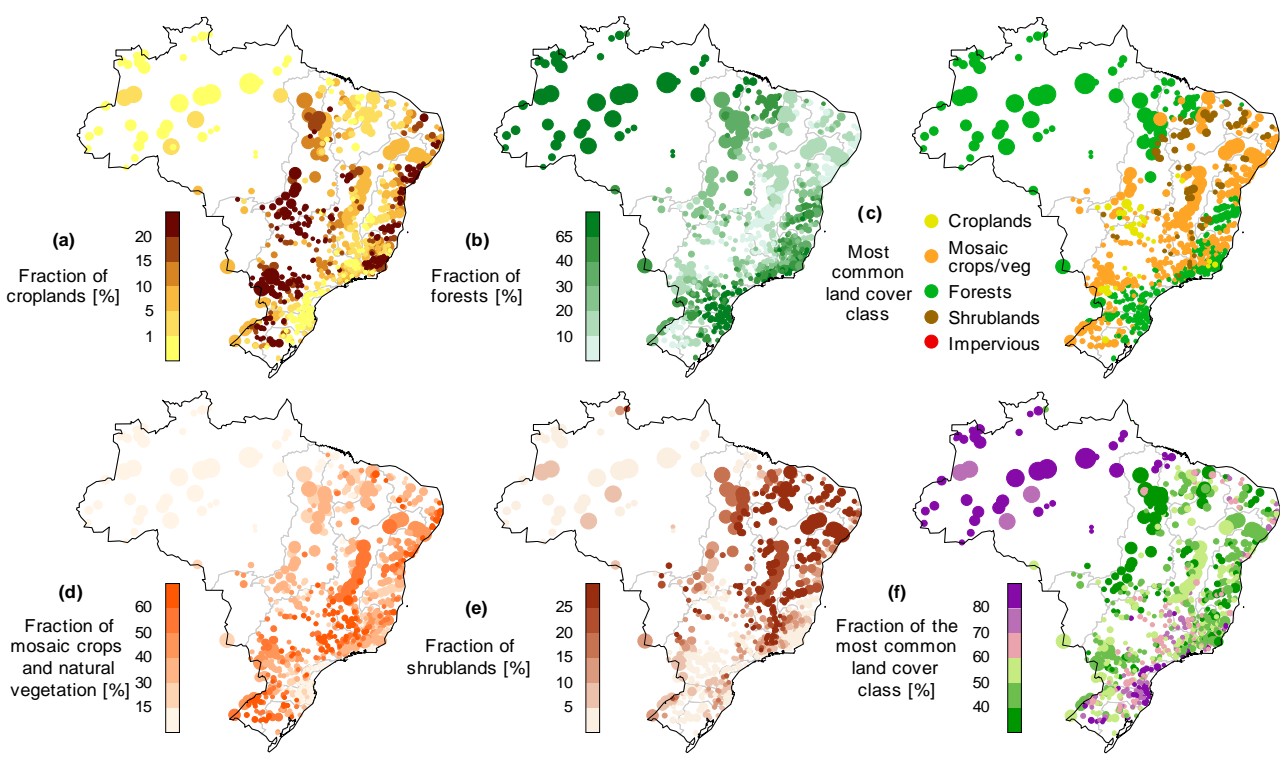




**Figure 7. Geologic characteristics. The size of the circles is proportional to the size of the catchment. The grey line indicates the limits of hydrographic regions.**





**Figure 8. Soil characteristics. The size of the circles is proportional to the size of the catchment. The grey line indicates the limits of hydrographic regions.**

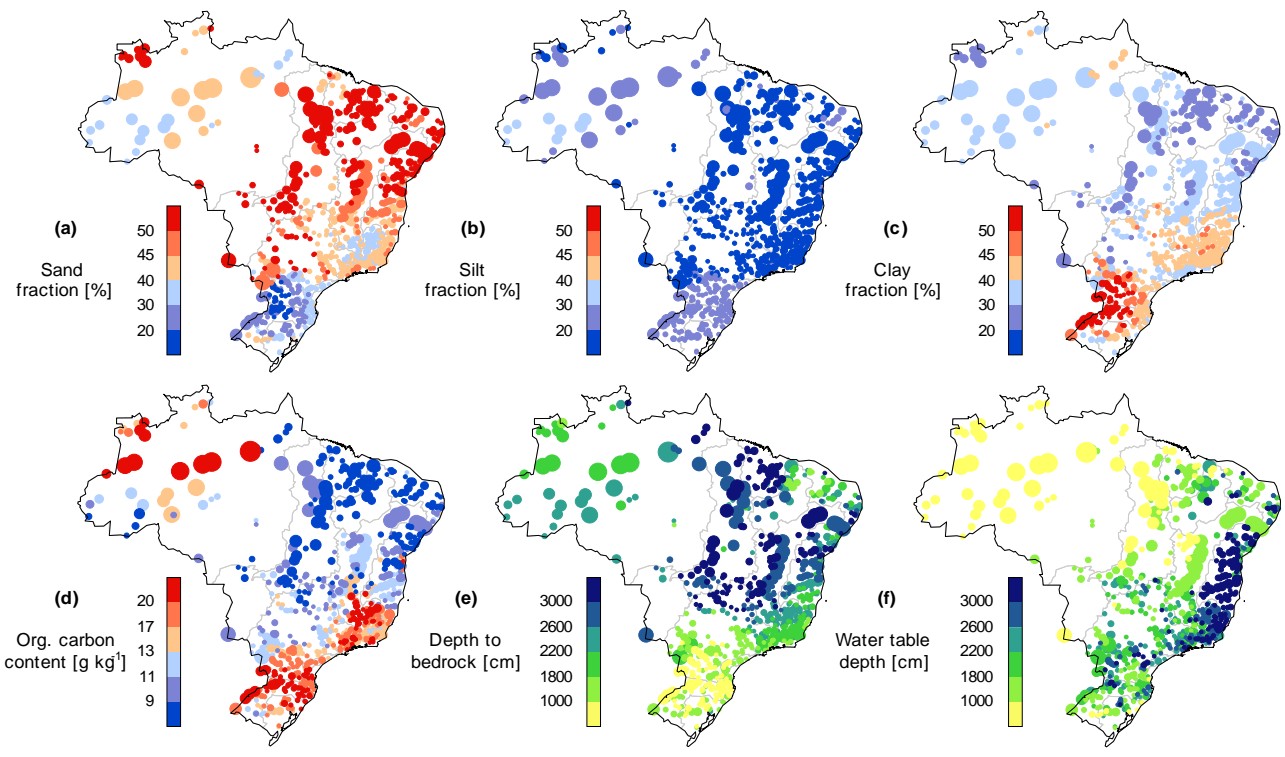



**Figure 9. Human intervention indices. The size of the circles is proportional to the size of the catchment. The grey line indicates the**
**limits of hydrographic regions.**

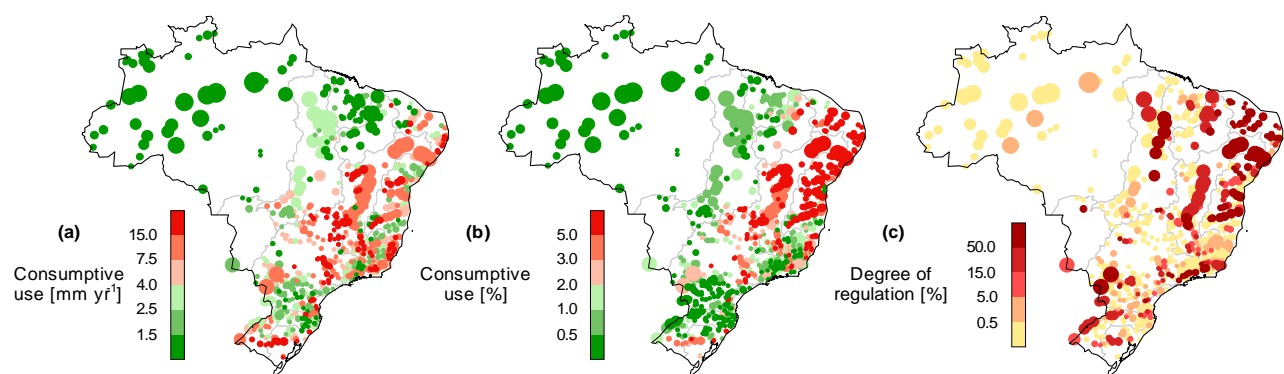