# Peer review of "CAMELS-BR: Hydrometeorological time series and landscape attributes for 897 catchments in Brazil"

_Earth System Science Data, 2020_

## Referee Comment (RC1) · Wouter Knoben (Referee) · 28 Apr 2020

Summary

The authors present a unified dataset for large-sample hydrology in Brazil. They provide raw streamflow data for 3713 gauges across Brazil, and catchment attributes for a subset of 897 catchments for which the authors consider the streamflow data to be of good quality. The catchment attributes are compiled from multiple sources and broadly cover topographic, climatic, hydrologic, land cover, geologic, soil and human intervention variables. The data are made publicly available through Zenodo.

[Figure]

First off, I would like to congratulate the authors on what must have been a massive amount of work. This looks like a tremendous dataset and it is clear that the authors must have spent considerable effort compiling it. From the success of the original CAMELS data and the more recent CAMELS-CL, it is clear that datasets such as this are very highly appreciated by the community. I think CAMELS-BR forms a strong addition to these two existing datasets. I have no major comments but compiled a list of smaller points that hopefully help the authors in clarifying a few things that are not entirely clear to me.

Review

L37: Suggest to change "validation" to "evaluation". See e.g. Oreskes et al. (1994)

L61: Is a word missing here? "... by institutions such as the ..."

L77-78: How do these numbers relate to the 897 catchments in the title?

L90: What are the native and new file formats?

L101: As far as I can tell CAMELS and CAMELS-CL cover the period 1989 to 2009 (at least). Why was the year 1989 not included in CAMELS-BR?

L108: Is this a complete list of quality control checks that were performed? The text "... errors such as ..." seems to imply that more checks were done but not listed here. A complete list of all quality control steps taken would be good (or rewriting the sentence without the words "such as" if the current list is already complete).

L117: To clarify, lines 117 to 121 only summarize the quality control by ANA? All 897 selected gauges have passed the authors' additional quality control described in lines 108-111?

L125: It might be good to expand the current meteorological indices with ae_mean (mean actual evaporation). This goes beyond what CAMELS and CAMELS-CL provide, but it might be a good way to remind the reader that actual evaporation data is

also provided.

L129: "no weight was applied if a cell is only partially covered by the catchment". Does this mean that partially covered cells are not used for calculating the catchment average or that all cells contribute to the average equally, whether the catchment fully covers them or not? Why was this particular choice made and can this be justified in some way?

L130-134: I don't fully understand the description of this limitation (maybe because of the previous comment). Does this mean that for some catchments and meteo products no data could be calculated?

L130-134: I think this limitation section can be stronger if the authors describe how they deal with this limitation during preparation of the data set.

L173-175: Are sine curves an appropriate representation of the temperature and precipitation regimes in Brazil? Was the accuracy of the calibrated sine curves comparable to the results in Berghuijs and Woods (2016)? Given how large the study area is, and that seasonality metrics tend to be somewhat specialized towards certain climate types, it might be useful to compute a few additional seasonality metrics (see e.g. Feng et al., 2019).

L207: is the Ladson digital filter the same approach as used in CAMELS and CAMELS-CL?

L378-388: It might be worthwhile to briefly discuss here what happens with consumptive water after it has been used. Does the predominantly evaporate/transpire or is it released back into the stream? In which way are the calculated streamflow indices affected by water use?

L408; "Lehner et al. (2011, Technical Documental)" Should this be "Technical Document"?

L441: "a new dataset comprising more than 3000 catchments in Brazil". It would be

helpful to add a line here to clarify that there is a subset of 897 basins, and which kind of data and attributes are available for the 3000+ and the 897 set.

Table 1: there is some inconsistency between time periods for various forcing variables. For consistency with CAMELS and CAMELS-CL, it would be nice if all variables are provided for 1979-2009.

Table 5: is "bare_frac" the same variable as "barren_frac" in CAMELS-CL? If so, it would be good to stick with consistent naming.

Figures: it is a bit difficult to make out any details in the figure in the south-east region, where gauge density is high. It might be worthwhile to not scale the data points according to catchment size (although keeping this scaling in Figure 1 is quite informative) in the data plots.

Figures: a follow-up suggestion to the previous comment is to add histograms to each data plot that summarize the information on the map (as was done in the original CAMELS paper). This makes it easier for the reader to see how the catchment attributes vary across their respective ranges.

Figures: I'm not sure whether a diverging colour scheme is very appropriate for continuous variables that have no clear breakpoint in the middle of the range. For example in Fig. 3a, I don't fully understand why catchments smaller than 5*10ˆ3 kmˆ2 are green and larger ones red. This implies some critical change between the smaller and larger catchments that I don't think is there. A continuous color scheme (e.g. Fig. 4d) would be more appropriate. This applies to multiple figures. Note, in cases such as Fig. 4c I think a diverging colour scheme is justified, because this makes it easier to distinguish positive and negative values.

Figure 4b: Do no aridity index values exceed 1.2?

Figure 4c: If I remember this metric correctly, values of -0.5 and +0.5 should be equivalent. Why do these values exceed -0.8 And +0.8?

[Figure]

References

Berghuijs, W. R., & Woods, R. A. (2016). A simple framework to quantitatively describe monthly precipitation and temperature climatology. International Journal of Climatology, 36, 3161–3174. https://doi.org/10.1002/joc.4544

Feng, X., Thompson, S. E., Woods, R., & Porporato, A. (2019). Quantifying asynchronicity of precipitation and potential evapotranspiration in Mediterranean climates. Geophysical Research Letters.https://doi.org/10.1029/2019GL085653

Oreskes, N., Shrader-Frechette, K., & Belitz, K. (1994). Verification, Validation, and Confirmation of Numerical Models in the Earth Sciences. Science, 263(5147), 641–646. https://doi.org/10.1126/science.263.5147.641

---

## Referee Comment (RC2) · Anonymous Referee #2 · 28 Apr 2020

This study produced a new public large-sample hydrological dataset for Brazil. It reports streamflow and meteo timeseries from thousands of gauges, and more detailed catchment descriptions for a subset of 867 catchments, including descriptions on hydrological signatures, geology, soils, human intervention, and landcover. Conveniently these descriptions follow standards defined by the previous CAMELS (US)and CAMELS-CL datasets. In addition, to solely providing the data, the sources of the data, and potential limitations and pitfalls are discussed.

The paper reads very well and descriptions are generally clear. The approach seems sound and the dataset will be of much value to the hydrology community (and beyond).

[Figure]

Therefore the paper seems very suitable for publication in ESSD. I want to thank the authors for writing this very useful paper and providing this valuable dataset to the community.

I only have very few minor points that could be clarified (in addition to minor comments posted in the first review by Knoben):

- To what extent do the ET estimates match P-Q when several years of data are available. This might be good to know, to get a first-order idea if the estimates seem somewhat reasonable.

- "The mean daily precipitation in Brazil is highest in the Amazon and in Southern Brazil, where it usually exceeds 5 mm day-1" I would replace "usually" to "on average" since the first is more often associated with a median than a mean.

- Figures often refer to "fractions" (which suggest 0-1) when instead "percentages" are displayed. Either is fine, but it would be nice if the use was consistent.
* * *

---

## Referee Comment (RC3) · Thibault Mathevet (Referee) · 7 Jun 2020

Summary :

The authors present an open-source hydrometeorological data sets in Brazil with ge-omorphological and human influences attributes. The authors followed the CAMELS standard of data-sets and complemented already existing data sets in USA, Chile and UK (soon). Building such a rich and usefull data sets is time and energy consuming, given the numerous sources of informations to gather, compile and summarize. I would like to thank the authors for making this data set open source and so documented, it will be very usefull for students, the water resources community in Brazil and the re-

searchers community worldwide. I really appreciate to see that CAMELS datasets become a standard, following Addor et al. (2019) recommandations. I hope that this standard will spread and that national datasets will be built worldwide.

In complement to the 2 previous reviews, I have very few comments. The manuscript is well structured and easy to read. I guess that the increasing use of CAMELS datasets might help to better assess the different sources of informations included. The Figures are very clear and give a good illustration of the information content of the datasets and spatial variability of indices. I particularly appreciated §9 Human intervention indices. These indices are particulary difficult to estimate. Even if they could be rather uncertain, these indices are at least a good basis to classify the level of human influences. These indices might be also usefull while working on Rainfall-Runoff model and understanding model performances/failures.

I clearly recommend this paper for publication in ESSD and encourage the community to use this data sets.

My few comments:

L227 : please clarify "The mean half-flow date".

To illustrate §4, 5 & 9, I encourage the authors to add a figure with Turc-Mezentsev water balance representation, with the runoff coefficient (Q/P) as a function of the humidity index (P/PET) (897 watersheds, 1990-2009 period). This figure would give a good representation of the water balance variability of the datasets, and the impact of some major human influences or uncertainties in the climatic/streamflow observations.

This datasets will probably be very usefull for Rainfall-Runoff model intercomparison studies (recently, Mathevet et al., 2020). In order to give a benchmark of hydrological model performances, I would encourage the authors to calibrate a commonly used conceptual Rainfall -Runoff model (such as GR4J model, freely available, Coron et al. 2017 or any other Rainfall-Runoff model). A very simple modeling framework might

gives the expected level of model performances on this datasets and the spatial variability of model performances. Providing such a benchmark could slightly improve the paper.

Add the number of watershed represented in the Figure captions (such as indicated in Table 1).

Is there a possibility to improve the density of watersheds in the western part of the country ? I understand that the spatial density of observations/stations is lower and that these stations might have been excluded for some reasons ? But, hypotheses of exclusion might be relaxed in regions where station density is lower, in order to have a more homogeneous spatial coverage of the county ?

Coron, L., Thirel, G., Delaigue, O., Perrin, C., Andréassian, V., 2017: The Suite of Lumped GR Hydrological Models in an R package, Environmental Modelling & Software, 94, 166-171, DOI: 10.1016/j.envsoft.2017.05.002.

Mathevet, T., Gupta, H., Perrin, C., Andréassian, V., Le Moine, N., 2020: Assessing the performance and robustness of two conceptual rainfall-runoff models on a worldwide sample of watersheds, Journal of Hydrology (2020), doi: 10.1016/j.jhydrol.2020.124698.

---

## Author Comment (AC1) · 16 Jul 2020

Reply to comments by Reviewer #1, Wouter Knoben.

We appreciate the helpful comments of Reviewer #1. The recommendations improved the clarity and the reproducibility of our work. Please, find below our reply to all the comments.

Comment #1: L37: Suggest to change "validation" to "evaluation". See e.g. Oreskes et al. (1994).

Reply: We have changed from "validation" to "evaluation" (line 38 in the track change

revised manuscript).

Comment #2: L61: Is a word missing here? "... by institutions such as the ..."

Reply: We have changed from "by institutions as the" to "by institutions such as the" (line 62 in the track change revised manuscript).

Comment #3: L77-78: How do these numbers relate to the 897 catchments in the title?

Reply: The 3,097 catchments mentioned in the manuscript were incorrect. We modified the sentence to refer to the 897 catchments in the title: "It includes daily streamflow time series from 3,679 stream gauges and, for a selected group of 897 catchments, daily meteorological time series and 65 catchment attributes from properties such as topography, climate, land cover, geology, soil, and human intervention." (lines 78-80 in the track change revised manuscript).

We removed from the dataset 34 catchments with available streamflow data, reducing the total number of gauges from 3,713 to 3,679. None of the 897 selected catchments were removed. Those 34 removed catchments are located outside Brazil and are not monitored by a Brazilian agency (such as the Brazilian National Water Agency), thus are outside the scope of CAMELS-BR. The total number of catchments were updated throughout the manuscript (lines 13, 89, 99, 103, and Table 1 of the track change revised manuscript).

Comment #4: L90: What are the native and new file formats?

Reply: We modified the sentence to specify the file formats: "Their values are unchanged but, to ease their processing, we converted the native files (i.e., Excel files with daily streamflows not disposed in chronological order) to a new file format (i.e., text files with daily streamflow in chronological order)." (lines 91-93 of the track change revised manuscript).

Comment #5: L101: As far as I can tell CAMELS and CAMELS-CL cover the period 1989 to 2009 (at least). Why was the year 1989 not included in CAMELS-BR?

Reply: CAMELS covers the water years from 1990 to 2009, which corresponds to 1 October 1989 to 30 September 2009. CAMELS-BR also covers the water years from 1990 to 2009, which corresponds to 1 September 1989 to 31 August 2009. So, they cover essentially the same period. This information was not clear in the manuscript, so we modified the sentence to "Firstly, we selected only gauges that have less than 5 % of missing streamflow data between the water years 1990 (starting on September 1, 1989) and 2009 (ending on August 31, 2009)." (lines 104-105 of the track change revised manuscript).

Comment #6: L108: Is this a complete list of quality control checks that were performed? The text "... errors such as ..." seems to imply that more checks were done but not listed here. A complete list of all quality control steps taken would be good (or rewriting the sentence without the words "such as" if the current list is already complete).

Reply: The current list of quality control checks is already complete. To make it clearer, we substituted "for errors such as" by "for the following errors" (line 113 of the track change revised manuscript).

Comment #7: L117: To clarify, lines 117 to 121 only summarize the quality control by ANA? All 897 selected gauges have passed the authors' additional quality control described in lines 108-111?

Reply: Yes, that summarizes only the quality control by ANA. To clarify, we modified the sentence to "To summarize the ANA metadata ..." (line 123 of the track change revised manuscript).

All 897 selected gauges have passed our additional quality control. To clarify, we modified the first sentence of the paragraph to "We individually screened the 897 selected streamflow time series for the following errors: ..." (line 113 of the track change revised manuscript).
Comment #8: L125: It might be good to expand the current meteorological indices with ae_mean (mean actual evaporation). This goes beyond what CAMELS and CAMELS-CL provide, but it might be a good way to remind the reader that actual evaporation data is also provided.

Reply: We added the climatic attribute et_mean (mean actual evapotranspiration) to the database and included its description in Table 3. Since another attribute was also added to the database (see Comment #12), we updated the total number of attributes from 63 to 65 throughout the manuscript; changed from 11 to 13 climatic attributes in Table 1; and modified the sentence "We computed thirteen climatic indices" (line 182 of the track change revised manuscript).

Comment #9: L129: "no weight was applied if a cell is only partially covered by the catchment". Does this mean that partially covered cells are not used for calculating the catchment average or that all cells contribute to the average equally, whether the catchment fully covers them or not? Why was this particular choice made and can this be justified in some way?

Reply: We modified the sentence to clarify it: "The daily values represent the average of all cells with their centroids intersected by the catchment, of which all cells contribute to the average equally, whether the catchment fully covers them or not. However, some catchments do not intersect the centroid of any cell. For those, we computed the daily values as the average of all cells partially covered by the catchment." (lines 134-138 of the track change revised manuscript). We chose this method because it is the most used in most algorithms (Tem alguma referência disso?).

Comment #10: L130-134: I don't fully understand the description of this limitation (maybe because of the previous comment). Does this mean that for some catchments and meteo products no data could be calculated?

Reply: Meteorological products were computed for all catchments, regardless of their sizes. We clarified the limitation of computing meteorological variables for catchments

smaller than a single cell by adding the sentence "This leads to the assumption that such a meteorological variable is homogeneous in catchments smaller than a single cell, even though this might not always be the case." (lines 140-141 of the track change revised manuscript).

Comment #11: L130-134: I think this limitation section can be stronger if the authors describe how they deal with this limitation during preparation of the data set.

Reply: Please refer to the reply to the two previous comments (Comment #10 and #11).

Comment #12: L173-175: Are sine curves an appropriate representation of the temperature and precipitation regimes in Brazil? Was the accuracy of the calibrated sine curves comparable to the results in Berghuijs and Woods (2016)? Given how large the study area is, and that seasonality metrics tend to be somewhat specialized towards certain climate types, it might be useful to compute a few additional seasonality metrics (see e.g. Feng et al., 2019).

We thank the reviewer for pointing out the paper by Feng et al. We extracted the asynchronicity index proposed by Feng et al. (2019) for each catchment and added this new index to CAMELS-BR.

We also added to the manuscript the following information:

(i) A description of the asynchronicity index on Table 3: "Asynchronicity between the annual precipitation and PET cycles, where high values represent high relative magnitude and phase differences".

(ii) "Those indices are complemented by the precipitation seasonality index (p_seasonality, Table 3), which relies on sine curves to approximate the monthly climatology of temperature and precipitation. While, for Brazil, the annual precipitation cycle is captured quite well, a sine curve provides a relatively rough approximation of the temperature cycle, particularly in the center of the country (around the state of

Goiás; Berghuijs and Woods, 2016). Hence, in addition to p_seasonality, we extracted the asynchronicity index proposed by Feng et al. (2019), which relies on information theory and has the advantage of being non-parametric (in particular, it does not assume sinusoidality)." (lines 190-196 of the track change revised manuscript).

(iii) "Northeastern Brazil (in particular, the states of Maranhão, Piauí, Ceará) has the highest values of asynchronicity index in the country (not shown), which corresponds to Mediterranean climates." (lines 204-206 of the track change revised manuscript).

References:

Berghuijs, W. R. and Woods, R. A.: A simple framework to quantitatively describe monthly precipitation and temperature climatology, Int. J. Climatol., 36(9), 3161–3174, doi:10.1002/joc.4544, 2016.

Feng, X., Thompson, S. E., Woods, R. and Porporato, A.: Quantifying Asynchronicity of Precipitation and Potential Evapotranspiration in Mediterranean Climates, Geophys. Res. Lett., 46(24), 14692–14701, doi:10.1029/2019GL085653, 2019.

Comment #13: L207: is the Ladson digital filter the same approach as used in CAMELS and CAMELS-CL?

Reply: Yes. All hydrological indices were computed using the same approach as in CAMELS and CAMELS-CL, including the usage of the Ladson digital filter. To clarify this, we added the sentence earlier in the same paragraph: "The hydrological signatures were computed in the same approach as in CAMELS, CAMEL-CL, and CAMELS-GB datasets." (lines 229-230 of the track change revised manuscript).

Comment #14: L378-388: It might be worthwhile to briefly discuss here what happens with consumptive water after it has been used. Does the predominantly evaporate/transpire or is it released back into the stream? In which way are the calculated streamflow indices affected by water use?

Reply: We added the definition of consumptive water use earlier in the section, when

it is mentioned for the first time: "Consumptive water use refers to water withdrawals that do not return to the catchment, for example, by evaporating, transpiring, or being incorporated into manufactured products." (lines 381-383 of the track change revised manuscript).

The streamflow indices are certainly affected by consumptive water use since it is an essential component of the water cycle (Milly et al., 2008; Hoekstra and Mekonnen, 2012). However, we do not know in which ways those indices are affected, as we do not know in exactly which ways the other ∼50 attributes affect the streamflow indices. This is an extensive and interesting topic of research (Montanari et al., 2012) and we hope that the CAMELS-BR dataset allows further investigations of what drives the hydrological behavior of catchments and in which way calculated streamflow indices are affected by water use.

References:

Hoekstra, A. Y. and Mekonnen, M. M.: The water footprint of humanity, Proceedings of the National Academy of Sciences, 109(9), 3232–3237, doi:10.1073/pnas.1109936109, 2012.

Milly, P. C. D., Betancourt, J., Falkenmark, M., Hirsch, R. M., Kundzewicz, Z. W., Lettenmaier, D. P. and Stouffer, R. J.: Stationarity Is Dead: Whither Water Management?, Science, 319(5863), 573–574, doi:10.1126/science.1151915, 2008.

Montanari, A., Young, G., Savenije, H. H. G., Hughes, D., Wagener, T., Ren, L. L., Koutsoyiannis, D., Cudennec, C., Toth, E., Grimaldi, S., Blöschl, G., Sivapalan, M., Beven, K., Gupta, H., Hipsey, M., Schaefli, B., Arheimer, B., Boegh, E., Schymanski, S. J., Di Baldassarre, G., Yu, B., Hubert, P., Huang, Y., Schumann, A., Post, D. A., Srinivasan, V., Harman, C., Thompson, S., Rogger, M., Viglione, A., McMillan, H., Characklis, G., Pang, Z. and Belyaev, V.: "Panta Rhei—Everything Flows": Change in hydrology and society—The IAHS Scientific Decade 2013–2022, Hydrological Sciences Journal, 58(6), 1256–1275, doi:10.1080/02626667.2013.809088, 2013.

[Figure]

Comment #15: L408; "Lehner et al. (2011, Technical Documental)" Should this be "Technical Document"?

Reply: We changed from "Technical Documental" to "Technical Document" (line 438 in the track change revised manuscript).

Comment #16: L441: "a new dataset comprising more than 3000 catchments in Brazil". It would be helpful to add a line here to clarify that there is a subset of 897 basins, and which kind of data and attributes are available for the 3000+ and the 897 set.

Reply: We clarified by modifying the sentence to "Here, we introduced the CAMELS-BR, a new dataset comprising streamflow time series for 3,679 catchments in Brazil and, for a selected quality-controlled set of 897 catchments, meteorological time series and 65 catchment attributes." (lines 470-473 of the track change revised manuscript).

Comment #17: Table 1: there is some inconsistency between time periods for various forcing variables. For consistency with CAMELS and CAMELS-CL, it would be nice if all variables are provided for 1979-2009.

Reply: Consistency between CAMELS datasets refers to the water years from 1990 to 2009. All attributes in CAMELS-BR cover this period and every time series includes at least those 20 years of data.

We have modified Table 1 to clarify the coverage period of each data source. Each data source covers distinct and non-coincidental periods. If we restricted all meteorological variables to include only coincident time periods, it would reduce the coverage periods from 1980-2018 to 1981-2014. We believe that it is more beneficial for the users of the dataset to include the entire cover period that each data source provides.

Comment #18: Table 5: is "bare_frac" the same variable as "barren_frac" in CAMELS-CL? If so, it would be good to stick with consistent naming.

Reply: Yes, both refer to the same variable. We renamed the variable from "bare_frac" to "barren_perc" because it refers to percentages instead of fractions (Table 5 of the

revised manuscript), as recommended by Comment #26 (Reviewer #2).

Comment #19: Figures: it is a bit difficult to make out any details in the figure in the south-east region, where gauge density is high. It might be worthwhile to not scale the data points according to catchment size (although keeping this scaling in Figure 1 is quite informative) in the data plots.

Reply: We have tried your suggestion and it seems that not scaling the symbols with catchment area did not solve this problem. It seems that it also hinders the visualization in low gauge density regions (see Fig R1c, f, i). Although the southeastern region has a high gauge density, those are usually the smallest catchments in the country (Fig R1a). We believe that scaling the symbols with catchment size facilitates the data visualization, particularly in the southeast.

We tried to improve the visualization by decreasing the size of all symbols but keeping them scaling with catchment size. In this way, it enhanced the visualization of high gauge density regions without hindering the visualization of low gauge density regions (Fig R1a-b, d-e, g-h). The following Figures were modified: Fig 1, 4, 5, 6, 7, 8, 9, and 10.

Comment #20: Figures: a follow-up suggestion to the previous comment is to add histograms to each data plot that summarize the information on the map (as was done in the original CAMELS paper). This makes it easier for the reader to see how the catchment attributes vary across their respective ranges.

Reply: We added a histogram to each map that shows a catchment attribute. The following figures were modified: Fig 1, 4, 5, 6, 7, 8, 9 and 10 in the revised manuscript.

Comment #21: Figures: I'm not sure whether a diverging colour scheme is very appropriate for continuous variables that have no clear breakpoint in the middle of the range. For example in Fig. 3a, I don't fully understand why catchments smaller than $5*10^3$ $km^2$ are green and larger ones red. This implies some critical change between the smaller and larger catchments that I don't think is there. A continuous color scheme (e.g. Fig. 4d) would be more appropriate. This applies to multiple figures. Note, in cases such as Fig. 4c I think a diverging colour scheme is justified, because this makes it easier to distinguish positive and negative values.

Reply: We changed to a sequential color scheme all Figures that had diverging color schemes and no clear breakpoint. The following Figures were modified: Fig 1c-d; Fig 4a and 4c; Fig 6c-f; Fig 7f; Fig 8c-d and f-g; Fig 9a-d; Fig 10a-b (of the revised manuscript).

Additionally, to improve visualization, we changed the classes of the color schemes of Fig 1a and 1c; Fig 4a and 4c; Fig 6f; Fig 8d; and Fig 10a (of the revised manuscript) and included additional gauges that were missing in Fig 1a.

Comment #22: Figure 4b: Do no aridity index values exceed 1.2?

Reply: The catchments in the most arid parts of Brazil frequently exceed aridity index values of 1.2. The color class with the largest values refers to aridity greater than 1.0 but not limited to 1.2. We believe it is clearer with the addition of histograms to represent the range of values in the figure (Fig 5 of the revised manuscript).

Comment #23: Figure 4c: If I remember this metric correctly, values of -0.5 and +0.5 should be equivalent. Why do these values exceed -0.8 And +0.8?

Reply: We imagine that the reviewer refers to another metric, because this metric, computed using Eq. 14 in Woods (2009), typically takes values between -1 (precipitation out of phase with temperature) and 1 (precipitation in phase with temperature, i.e. simultaneous peaks), while values close to 0 indicate uniform precipitation throughout the year. Hence, values smaller than -0.8 or greater than 0.8 are possible, and values of -0.5 and +0.5 are not equivalent. We have clarified the description of this metric in Table 3, added the references to this equation, and added the equations used to compute other indices in Table 4.

References:

Woods, R. A.: Analytical model of seasonal climate impacts on snow hydrology: Continuous snowpacks, Adv. Water Resour., 32, 1465–1481, https://doi.org/10.1016/j.advwatres.2009.06.011, 2009.

Reply to comments by Reviewer #2.

We appreciate the helpful comments of Reviewer #2. We have agreed to all recommendations. Please, find below our reply to all the comments.

Comment #24: To what extent do the ET estimates match P-Q when several years of data are available. This might be good to know, to get a first-order idea if the estimates seem somewhat reasonable.

Reply: To analyze to what extent ET estimates matches P-Q, we added a new scatterplot with the long-term water balance (Fig. 3a-b in the revised manuscript).

The following paragraph was added to describe the conclusions from the figure: "The long-term water balance is accurate for most catchments, using either the estimated evapotranspiration from GLEAM (Fig. 3a) or MGB (Fig. 3b). Both evapotranspiration data sources indicate that the highest data uncertainties occur in the Amazon and smaller catchments in the Paraná and the Southeastern Atlantic regions, since those catchments are further away from the 1:1 line in Fig. 3a-b. The same conclusions are derived from the runoff coefficient as a function of the humidity index (Fig. 3c). In addition, there are remarkable differences between GLEAM and MGB estimates, where evapotranspiration from GLEAM is substantially higher in the Amazon basin and substantially lower in the Eastern and the Western NE Atlantic regions." (lines 153-159 of the track change revised manuscript).

Comment #25: "The mean daily precipitation in Brazil is highest in the Amazon and in Southern Brazil, where it usually exceeds 5 mm day-1" I would replace "usually" to "on average" since the first is more often associated with a median than a mean.

Reply: We changed from "usually" to "on average" (line 202 in the track change revised manuscript).

Comment #26: Figures often refer to "fractions" (which suggest 0-1) when instead "percentages" are displayed. Either is fine, but it would be nice if the use was consistent.

Reply: We consistently modified all attribute names and values to refer to percentages instead of fractions. We changed the names (from "_frac" to "_perc") and the descriptions of attributes in the following: Tables 2, 5, 6, and 7; Figures 1, 7, 8, and 9; and line 123 of the track change revised manuscript.

Additionally, to maintain consistency across CAMELS datasets, we have changed the names of the following attributes: bedrock_depth, reservoirs_vol, regulation_degree, consumptive_use, and consumptive_use_perc (Tables 7 and 8).

Reply to comments by Reviewer #3, Thibault Mathevet.

We appreciate the helpful comments of Reviewer #3. The recommendations improved the clarity and the reproducibility of our work. Please, find below our reply to all the comments.

Comment #27: L227 : please clarify "The mean half-flow date".

Reply: We modified the sentence to "The mean half-flow date (i.e., when the cumulative discharge since 1st September reaches half of the annual discharge) ..." (line 253 of the track change revised manuscript).

Comment #28: To illustrate §4, 5 & 9, I encourage the authors to add a figure with Turc-Mezentsev water balance representation, with the runoff coefficient (Q/P) as a function of the humidity index (P/PET) (897 watersheds, 1990-2009 period). This figure would give a good representation of the water balance variability of the datasets, and the impact of some major human influences or uncertainties in the climatic/streamflow observations.

Reply: To analyze the variability of the water balance, we added a new scatterplot with the runoff coefficient as a function of the humidity index (Fig. 3c in the revised manuscript).

The following paragraph was added to describe the conclusions from the figure (as mentioned in Comment #24 from Reviewer #2): "The long-term water balance is accurate for most catchments, using either the estimated evapotranspiration from GLEAM (Fig. 3a) or MGB (Fig. 3b). Both evapotranspiration data sources indicate that the highest data uncertainties occur in the Amazon and smaller catchments in the Paraná and the Southeastern Atlantic regions, since those catchments are further away from the 1:1 line in Fig. 3a-b. The same conclusions are derived from visualizing the runoff coefficient as a function of the humidity index (Fig. 3c). In addition, there are remarkable differences between GLEAM and MGB estimates, where evapotranspiration from GLEAM is substantially higher in the Amazon basin and substantially lower in the Eastern and the Western NE Atlantic regions." (lines 153-159 of the track-change revised manuscript).

Comment #29: This datasets will probably be very usefull for Rainfall-Runoff model intercomparison studies (recently, Mathevet et al., 2020). In order to give a benchmark of hydrological model performances, I would encourage the authors to calibrate a commonly used conceptual Rainfall -Runoff model (such as GR4J model, freely available, Coron et al. 2017 or any other Rainfall-Runoff model). A very simple modeling framework might gives the expected level of model performances on this datasets and the spatial variability of model performances. Providing such a benchmark could slightly improve the paper.

Reply: We agree that providing a set of hydrological simulations to be used as a benchmark is very valuable. We will provide in the database the streamflow simulation for a set of approximately 500 of the catchments. Those simulations are extracted from Siqueira et al. (2018) that used a fully coupled hydrologic–hydrodynamic model (MGB; Modelo Hidrológico de Grandes Bacias) to the continental domain of South America.

While calibrating and analyzing a different rainfall-runoff model would be very valuable, we believe that we are already covering a lot of ground in this paper, by providing a wide range of time series and catchment attributes for a country in which such a dataset does not exist yet. The data processing to produce CAMELS-BR took more than two years and we are concerned that properly setting up another hydrological model (even a simple one) and analyzing its simulations for hundreds of catchments would add a further delay. It is our intention to keep adding to CAMELS-BR and we anticipate that hydrological simulations for these catchments using different models will be produced in the near future and be shared with the community.

References:

Siqueira, V. A., Paiva, R. C. D., Fleischmann, A. S., Fan, F. M., Ruhoff, A. L., Pontes, P. R. M., Paris, A., Calmant, S. and Collischonn, W.: Toward continental hydrologic–hydrodynamic modeling in South America, Hydrol. Earth Syst. Sci., 22(9), 4815–4842, doi:10.5194/hess-22-4815-2018, 2018.

Comment #30: Add the number of watershed represented in the Figure captions (such as indicated in Table 1).

Reply: We added the number of catchments in the captions of Figures 1, 4, 5, 6, 7, 8, 9, and 10 of the revised manuscript.

Comment #31: Is there a possibility to improve the density of watersheds in the western part of the country ? I understand that the spatial density of observations/stations is lower and that these stations might have been excluded for some reasons ? But, hypotheses of exclusion might be relaxed in regions where station density is lower, in order to have a more homogeneous spatial coverage of the county ?

Reply: Even if we use a more relaxed selection criterion, the gauge density increase would be noticeable only in a small portion of the western part of the country (see Fig. R2). If we consider all basins with 10 years of data and less than 5% missing (from

2000-2009), only the upper Paraguay basin would have increased gauge density (Fig. R2c). Changes in other regions, such as the Amazon, Tocantins-Araguaia, and lower Paraguay, would barely be noticeable.

Changing to a more relaxed selection criterion would remove the consistency among CAMELS-BR and the other CAMELS datasets. Additionally, reducing the coverage period would lead to substantial increases in the uncertainty of climatic and hydrological indices. We are aware that the users of the dataset might want to consider those additional catchments in further research, which is why we have included streamflow data from all 3679 catchments in Brazil in the manuscript.

**(a)** Symbols scaling with basin area
(previous manuscript)

**(b)** Symbols scaling with basin area
(revised manuscript)

**(c)** Symbols not scaling with basin area

Catchment
area [km$^2$]

$10^6$
$10^5$
$10^4$
$10^3$

**(d)**  **(e)**  **(f)**

Runoff
ratio [-]

0.55
0.45
0.35
0.25
0.15

**(g)**  **(h)**  **(i)**

Most common
land cover
class

● Croplands
● Mosaic
crops/veg
● Forests
● Shrublands
● Impervious

**Fig. 1.** Figure R1. Example of attributes with symbols scaling with catchment size in the previous manuscript, in the revised manuscript, and with symbols not scaling with catchment size.

[Figure]

(a)

20 years of data
with less than 5%
missing (1990-2009)

(b)

15 years of data
with less than 5%
missing (1995-2009)

(c)

10 years of data
with less than 5%
missing (2000-2009)

**Fig. 2.** Figure R2: Gauges selected using three different criteria.

---

## Author Response (AR1)

**Reply to comments by Reviewer #1, Wouter Knoben.**

We appreciate the helpful comments of Reviewer #1. The recommendations improved the clarity and the reproducibility of our work. Please, find below our reply to all the comments.

For clarity, the authors' responses are inserted as blue text.

Comment #1: L37: Suggest to change "validation" to "evaluation". See e.g. Oreskes et al. (1994).

Reply: We have changed from "validation" to "evaluation" (line 38 in the track change revised manuscript).

Comment #2: L61: Is a word missing here? "… by institutions such as the …"

Reply: We have changed from "by institutions as the" to "by institutions such as the" (line 62 in the track change revised manuscript).

Comment #3: L77-78: How do these numbers relate to the 897 catchments in the title?

Reply: The 3,097 catchments mentioned in the manuscript were incorrect. We modified the sentence to refer to the 897 catchments in the title: "It includes daily streamflow time series from 3,679 stream gauges and, for a selected group of 897 catchments, daily meteorological time series and 65 catchment attributes from properties such as topography, climate, land cover, geology, soil, and human intervention." (lines 78-80 in the track change revised manuscript).

We removed from the dataset 34 catchments with available streamflow data, reducing the total number of gauges from 3,713 to 3,679. None of the 897 selected catchments were removed. Those 34 removed catchments are located outside Brazil and are not monitored by a Brazilian agency (such as the Brazilian National Water Agency), thus are outside the scope of CAMELS-BR. The total number of catchments were updated throughout the manuscript (lines 13, 89, 99, 103, and Table 1 of the track change revised manuscript).

Comment #4: L90: What are the native and new file formats?

Reply: We modified the sentence to specify the file formats: "Their values are unchanged but, to ease their processing, we converted the native files (i.e., Excel files with daily streamflows not disposed in chronological order) to a new file format (i.e., text files with daily streamflow in chronological order)." (lines 91-93 of the track change revised manuscript).

Comment #5: L101: As far as I can tell CAMELS and CAMELS-CL cover the period 1989 to 2009 (at least). Why was the year 1989 not included in CAMELS-BR?

Reply: CAMELS covers the water years from 1990 to 2009, which corresponds to 1 October 1989 to 30 September 2009. CAMELS-BR also covers the water years from 1990 to 2009, which corresponds to 1 September 1989 to 31 August 2009. So, they cover essentially the same period.

This information was not clear in the manuscript, so we modified the sentence to "Firstly, we selected only gauges that have less than 5 % of missing streamflow data between the water years 1990 (starting on September 1, 1989) and 2009 (ending on August 31, 2009)." (lines 104-105 of the track change revised manuscript).

Comment #6: L108: Is this a complete list of quality control checks that were performed? The text "… errors such as …" seems to imply that more checks were done but not listed here. A complete list of all quality control steps taken would be good (or rewriting the sentence without the words "such as" if the current list is already complete).

Reply: The current list of quality control checks is already complete. To make it clearer, we substituted "for errors such as" by "for the following errors" (line 113 of the track change revised manuscript).

Comment #7: L117: To clarify, lines 117 to 121 only summarize the quality control by ANA? All 897 selected gauges have passed the authors' additional quality control described in lines 108-111?

Reply: Yes, that summarizes only the quality control by ANA. To clarify, we modified the sentence to "To summarize the ANA metadata …" (line 123 of the track change revised manuscript).
All 897 selected gauges have passed our additional quality control. To clarify, we modified the first sentence of the paragraph to "We individually screened the 897 selected streamflow time series for the following errors: …" (line 113 of the track change revised manuscript).

Comment #8: L125: It might be good to expand the current meteorological indices with ae_mean (mean actual evaporation). This goes beyond what CAMELS and CAMELS-CL provide, but it might be a good way to remind the reader that actual evaporation data is also provided.

Reply: We added the climatic attribute et_mean (mean actual evapotranspiration) to the database and included its description in Table 3. Since another attribute was also added to the database (see Comment #12), we updated the total number of attributes from 63 to 65 throughout the manuscript; changed from 11 to 13 climatic attributes in Table 1; and modified the sentence "We computed thirteen climatic indices" (line 182 of the track change revised manuscript).

Comment #9: L129: "no weight was applied if a cell is only partially covered by the catchment". Does this mean that partially covered cells are not used for calculating the catchment average or that all cells contribute to the average equally, whether the catchment fully covers them or not? Why was this particular choice made and can this be justified in some way?

Reply: We modified the sentence to clarify it: "The daily values represent the average of all cells with their centroids intersected by the catchment, of which all cells contribute to the average equally, whether the catchment fully covers them or not. However, some catchments do not intersect the centroid of any cell. For those, we computed the daily values as the average of all cells partially covered by the

catchment." (lines 134-138 of the track change revised manuscript). We chose this method because it is the most used in most algorithms.

Comment #10: L130-134: I don't fully understand the description of this limitation (maybe because of the previous comment). Does this mean that for some catchments and meteo products no data could be calculated?

Reply: Meteorological products were computed for all catchments, regardless of their sizes. We clarified the limitation of computing meteorological variables for catchments smaller than a single cell by adding the sentence "This leads to the assumption that such a meteorological variable is homogeneous in catchments smaller than a single cell, even though this might not always be the case." (lines 140-141 of the track change revised manuscript).

Comment #11: L130-134: I think this limitation section can be stronger if the authors describe how they deal with this limitation during preparation of the data set.

Reply: Please refer to the reply to the two previous comments (Comment #10 and #11).

Comment #12: L173-175: Are sine curves an appropriate representation of the temperature and precipitation regimes in Brazil? Was the accuracy of the calibrated sine curves comparable to the results in Berghuijs and Woods (2016)? Given how large the study area is, and that seasonality metrics tend to be somewhat specialized towards certain climate types, it might be useful to compute a few additional seasonality metrics (see e.g. Feng et al., 2019).

We thank the reviewer for pointing out the paper by Feng et al. We extracted the asynchronicity index proposed by Feng et al. (2019) for each catchment and added this new index to CAMELS-BR.
We also added to the manuscript the following information:
(i) A description of the asynchronicity index on Table 3: "Asynchronicity between the annual precipitation and PET cycles, where high values represent high relative magnitude and phase differences".
(ii) "Those indices are complemented by the precipitation seasonality index (p_seasonality, Table 3), which relies on sine curves to approximate the monthly climatology of temperature and precipitation. While, for Brazil, the annual precipitation cycle is captured quite well, a sine curve provides a relatively rough approximation of the temperature cycle, particularly in the center of the country (around the state of Goiás; Berghuijs and Woods, 2016). Hence, in addition to p_seasonality, we extracted the asynchronicity index proposed by Feng et al. (2019), which relies on information theory and has the advantage of being non-parametric (in particular, it does not assume sinusoidality)." (lines 190-196 of the track change revised manuscript)
(iii) "Northeastern Brazil (in particular, the states of Maranhão, Piauí, Ceará) has the highest values of asynchronicity index in the country (not shown), which corresponds to Mediterranean climates." (lines 204-206 of the track change revised manuscript).

References:

Berghuijs, W. R. and Woods, R. A.: A simple framework to quantitatively describe monthly precipitation and temperature climatology, Int. J. Climatol., 36(9), 3161–3174, doi:10.1002/joc.4544, 2016.

Feng, X., Thompson, S. E., Woods, R. and Porporato, A.: Quantifying Asynchronicity of Precipitation and Potential Evapotranspiration in Mediterranean Climates, Geophys. Res. Lett., 46(24), 14692–14701, doi:10.1029/2019GL085653, 2019.

Comment #13: L207: is the Ladson digital filter the same approach as used in CAMELS and CAMELS-CL?

Reply: Yes. All hydrological indices were computed using the same approach as in CAMELS and CAMELS-CL, including the usage of the Ladson digital filter. To clarify this, we added the sentence earlier in the same paragraph: "The hydrological signatures were computed in the same approach as in CAMELS, CAMEL-CL, and CAMELS-GB datasets." (lines 229-230 of the track change revised manuscript).

Comment #14: L378-388: It might be worthwhile to briefly discuss here what happens with consumptive water after it has been used. Does the predominantly evaporate/transpire or is it released back into the stream? In which way are the calculated streamflow indices affected by water use?

Reply: We added the definition of consumptive water use earlier in the section, when it is mentioned for the first time: "Consumptive water use refers to water withdrawals that do not return to the catchment, for example, by evaporating, transpiring, or being incorporated into manufactured products." (lines 381-383 of the track change revised manuscript).

The streamflow indices are certainly affected by consumptive water use since it is an essential component of the water cycle (Milly et al., 2008; Hoekstra and Mekonnen, 2012). However, we do not know in which ways those indices are affected, as we do not know in exactly which ways the other ~50 attributes affect the streamflow indices. This is an extensive and interesting topic of research (Montanari et al., 2012) and we hope that the CAMELS-BR dataset allows further investigations of what drives the hydrological behavior of catchments and in which way calculated streamflow indices are affected by water use.

Comment #18: Table 5: is "bare_frac" the same variable as "barren_frac" in CAMELS-CL? If so, it would be good to stick with consistent naming.

Reply: Yes, both refer to the same variable. We renamed the variable from "bare_frac" to "barren_perc" because it refers to percentages instead of fractions (Table 5 of the revised manuscript), as recommended by Comment #26 (Reviewer #2).

Comment #19: Figures: it is a bit difficult to make out any details in the figure in the south-east region, where gauge density is high. It might be worthwhile to not scale the data points according to catchment size (although keeping this scaling in Figure 1 is quite informative) in the data plots.

Reply: We have tried your suggestion and it seems that not scaling the symbols with catchment area did not solve this problem. It seems that it also hinders the visualization in low gauge density regions (see Fig R1c, f, i). Although the southeastern region has a high gauge density, those are usually the smallest catchments in the country

(Fig R1a). We believe that scaling the symbols with catchment size facilitates the data visualization, particularly in the southeast.

We tried to improve the visualization by decreasing the size of all symbols but keeping them scaling with catchment size. In this way, it enhanced the visualization of high gauge density regions without hindering the visualization of low gauge density regions (Fig R1a-b, d-e, g-h). The following Figures were modified: Fig 1, 4, 5, 6, 7, 8, 9, and 10.

**Figure R1. Example of attributes with symbols scaling with catchment size in the previous manuscript (a, d, g), in the revised manuscript (b, e, h), and with symbols not scaling with catchment size (c, f, i).**

[Figure]

Comment #20: Figures: a follow-up suggestion to the previous comment is to add histograms to each data plot that summarize the information on the map (as was done in the original CAMELS paper). This makes it easier for the reader to see how the catchment attributes vary across their respective ranges.

Reply: We added a histogram to each map that shows a catchment attribute. The following figures were modified: Fig 1, 4, 5, 6, 7, 8, 9 and 10 in the revised manuscript.

Comment #21: Figures: I'm not sure whether a diverging colour scheme is very appropriate for continuous variables that have no clear breakpoint in the middle of the

range. For example in Fig. 3a, I don't fully understand why catchments smaller than 5*10^3 km^2 are green and larger ones red. This implies some critical change between the smaller and larger catchments that I don't think is there. A continuous color scheme (e.g. Fig. 4d) would be more appropriate. This applies to multiple figures. Note, in cases such as Fig. 4c I think a diverging colour scheme is justified, because this makes it easier to distinguish positive and negative values.

Reply: We changed to a sequential color scheme all Figures that had diverging color schemes and no clear breakpoint. The following Figures were modified: Fig 1c-d; Fig 4a and 4c; Fig 6c-f; Fig 7f; Fig 8c-d and f-g; Fig 9a-d; Fig 10a-b (of the revised manuscript).

Additionally, to improve visualization, we changed the classes of the color schemes of Fig 1a and 1c; Fig 4a and 4c; Fig 6f; Fig 8d; and Fig 10a (of the revised manuscript) and included additional gauges that were missing in Fig 1a.

Comment #22: Figure 4b: Do no aridity index values exceed 1.2?

Reply: The catchments in the most arid parts of Brazil frequently exceed aridity index values of 1.2. The color class with the largest values refers to aridity greater than 1.0 but not limited to 1.2. We believe it is clearer with the addition of histograms to represent the range of values in the figure (Fig 5 of the revised manuscript).

Comment #23: Figure 4c: If I remember this metric correctly, values of -0.5 and +0.5 should be equivalent. Why do these values exceed -0.8 And +0.8?

Reply: We imagine that the reviewer refers to another metric, because this metric, computed using Eq. 14 in Woods (2009), typically takes values between -1 (precipitation out of phase with temperature) and 1 (precipitation in phase with temperature, i.e. simultaneous peaks), while values close to 0 indicate uniform precipitation throughout the year. Hence, values smaller than -0.8 or greater than 0.8 are possible, and values of -0.5 and +0.5 are not equivalent. We have clarified the description of this metric in Table 3, added the references to this equation, and added the equations used to compute other indices in Table 4.

References:
Woods, R. A.: Analytical model of seasonal climate impacts on snow hydrology: Continuous snowpacks, Adv. Water Resour., 32, 1465–1481, https://doi.org/10.1016/j.advwatres.2009.06.011, 2009.

**Reply to comments by Reviewer #2.**

We appreciate the helpful comments of Reviewer #2. We have agreed to all recommendations. Please, find below our reply to all the comments.
For clarity, the authors' responses are inserted as blue text.

Comment #24: To what extent do the ET estimates match P-Q when several years of data are available. This might be good to know, to get a first-order idea if the estimates seem somewhat reasonable.

Reply: To analyze to what extent ET estimates matches P-Q, we added a new scatterplot with the long-term water balance (Fig. 3a-b in the revised manuscript).
The following paragraph was added to describe the conclusions from the figure: "The long-term water balance is accurate for most catchments, using either the estimated evapotranspiration from GLEAM (Fig. 3a) or MGB (Fig. 3b). Both evapotranspiration data sources indicate that the highest data uncertainties occur in the Amazon and smaller catchments in the Paraná and the Southeastern Atlantic regions, since those catchments are further away from the 1:1 line in Fig. 3a-b. The same conclusions are derived from the runoff coefficient as a function of the humidity index (Fig. 3c). In addition, there are remarkable differences between GLEAM and MGB estimates, where evapotranspiration from GLEAM is substantially higher in the Amazon basin and substantially lower in the Eastern and the Western NE Atlantic regions." (lines 153-159 of the track change revised manuscript).

Comment #25: "The mean daily precipitation in Brazil is highest in the Amazon and in Southern Brazil, where it usually exceeds 5 mm day-1" I would replace "usually" to "on average" since the first is more often associated with a median than a mean.

Reply: We changed from "usually" to "on average" (line 202 in the track change revised manuscript).

Comment #26: Figures often refer to "fractions" (which suggest 0-1) when instead "percentages" are displayed. Either is fine, but it would be nice if the use was consistent.

Reply: We consistently modified all attribute names and values to refer to percentages instead of fractions. We changed the names (from "_frac" to "_perc") and the descriptions of attributes in the following: Tables 2, 5, 6, and 7; Figures 1, 7, 8, and 9; and line 123 of the track change revised manuscript.
Additionally, to maintain consistency across CAMELS datasets, we have changed the names of the following attributes: bedrock_depth, reservoirs_vol, regulation_degree, consumptive_use, and consumptive_use_perc (Tables 7 and 8).

**Reply to comments by Reviewer #3, Thibault Mathevet.**

We appreciate the helpful comments of Reviewer #3. The recommendations improved the clarity and the reproducibility of our work. Please, find below our reply to all the comments.

For clarity, the authors' responses are inserted as blue text.

Comment #27: L227 : please clarify "The mean half-flow date".

Reply: We modified the sentence to "The mean half-flow date (i.e., when the cumulative discharge since 1st September reaches half of the annual discharge) …" (line 253 of the track change revised manuscript).

Comment #28: To illustrate §4, 5 & 9, I encourage the authors to add a figure with Turc-Mezentsev water balance representation, with the runoff coefficient (Q/P) as a function of the humidity index (P/PET) (897 watersheds, 1990-2009 period). This figure would give a good representation of the water balance variability of the datasets, and the impact of some major human influences or uncertainties in the climatic/streamflow observations.

Reply: To analyze the variability of the water balance, we added a new scatterplot with the runoff coefficient as a function of the humidity index (Fig. 3c in the revised manuscript).

The following paragraph was added to describe the conclusions from the figure (as mentioned in Comment #24 from Reviewer #2): "The long-term water balance is accurate for most catchments, using either the estimated evapotranspiration from GLEAM (Fig. 3a) or MGB (Fig. 3b). Both evapotranspiration data sources indicate that the highest data uncertainties occur in the Amazon and smaller catchments in the Paraná and the Southeastern Atlantic regions, since those catchments are further away from the 1:1 line in Fig. 3a-b. The same conclusions are derived from visualizing the runoff coefficient as a function of the humidity index (Fig. 3c). In addition, there are remarkable differences between GLEAM and MGB estimates, where evapotranspiration from GLEAM is substantially higher in the Amazon basin and substantially lower in the Eastern and the Western NE Atlantic regions." (lines 153-159 of the track-change revised manuscript).

Comment #29: This datasets will probably be very usefull for Rainfall-Runoff model intercomparison studies (recently, Mathevet et al., 2020). In order to give a benchmark of hydrological model performances, I would encourage the authors to calibrate a commonly used conceptual Rainfall -Runoff model (such as GR4J model, freely available, Coron et al. 2017 or any other Rainfall-Runoff model). A very simple modeling framework might gives the expected level of model performances on this datasets and the spatial variability of model performances. Providing such a benchmark could slightly improve the paper.

Reply: We agree that providing a set of hydrological simulations to be used as a benchmark is very valuable. We will provide in the database the streamflow simulation for a set of approximately 500 of the catchments. Those simulations are extracted from Siqueira et al. (2018) that used a fully coupled hydrologic–hydrodynamic model (MGB; Modelo Hidrológico de Grandes Bacias) to the continental domain of South America.

While calibrating and analyzing a different rainfall-runoff model would be very valuable, we believe that we are already covering a lot of ground in this paper, by providing a wide range of time series and catchment attributes for a country in which such a dataset does not exist yet. The data processing to produce CAMELS-BR took more than two years and we are concerned that properly setting up another hydrological model (even a simple one) and analyzing its simulations for hundreds of catchments would add a further delay. It is our intention to keep adding to CAMELS-BR and we anticipate that hydrological simulations for these catchments using different models will be produced in the near future and be shared with the community.

Changing to a more relaxed selection criterion would remove the consistency among CAMELS-BR and the other CAMELS datasets. Additionally, reducing the coverage period would lead to substantial increases in the uncertainty of climatic and hydrological indices. We are aware that the users of the dataset might want to consider those additional catchments in further research, which is why we have included streamflow data from all 3679 catchments in Brazil in the manuscript.

**Figure R2: Gauges selected using three different criteria.**

[Figure]

(a)

20 years of data
with less than 5%
missing (1990-2009)

(b)

15 years of data
with less than 5%
missing (1995-2009)

(c)

10 years of data
with less than 5%
missing (2000-2009)

[revised manuscript text omitted]
, to ease their processing, we converted the native files (i.e., Excel files with daily streamflows not disposed in chronological order) to a new file format (i.e., text files with daily streamflow in chronological order). we provide these time series in a different file format to ease their processing. ANA estimates daily streamflow either by (i) taking two daily stream stage measurements, one in the morning (at 7 am) and another in the afternoon (at 5 pm), which are averaged and transformed into discharge using a stage-discharge relationship (rating curve); or (ii) resorting to regionalization methods when no stream stage measurements are available (no further details on the methods are provided by ANA). The raw streamflow time series cover different periods, ranging from a few days to

more than a century. Additionally, although ANA performs data quality checks, these time series include inconsistencies such as typographical errors and days with missing data. The  3679 gauges are irregularly distributed throughout the country

100 (Fig. 1a). Overall, their spatial distribution is denser and their time series longer in the Southern Atlantic, Southeastern Atlantic, and Paraná hydrographic regions (Fig. 1a and 1b).

The second set of streamflow time series includes 897 gauges, and here we simply refer to them as "streamflow" time series (Table 1). This is the set of gauges used to compute the catchment attributes. It is a subset from the previous  3679 gauges, which resulted from two selection criteria. Firstly, we selected only gauges that have less than 5 % of missing streamflow data

105 between the water years  1990 (starting on September 1, 1989) and 2009 (ending on August 31, 2009). We chose the water years from 1990 to 2009 because (i) it is the period with the largest number of stream gauges with available data (Fig. 2), and (ii) it coincides with the period of analysis from other CAMELS datasets (Addor et al., 2017; Alvarez-Garreton et al., 2018), allowing for direct comparisons with them. Secondly, we only considered catchments for which boundaries have been delimited by Do et al. (2018) and for which there is a good match with the area estimated by the data

110 provider (see Sect. 3). Although the hydrological signatures introduced below were computed using data from 1990 to 2009, the time series for the 897 stream gauges include data from 1980 to 2018 when available, to enable complementary analyses by other users.

We individually screened the 897 selected streamflow time series between 1990 and 2009 for the following errors:  zeroes or repeated values instead of missing values, abrupt changes resulting from changes in measurement instruments or

115 rating curves, annual streamflow larger than annual precipitation, and unrealistic daily streamflow values (i.e., larger than 1,000 mm day$^{-1}$). Gauges affected by such errors were not included in the set of 897 catchments. In addition, we summarized the streamflow metadata provided by ANA as follows. For each daily streamflow measurement, we provide two pieces of information (Table 1). The first metadata variable, "qual_control_by_ana", was set to 1 if the data was quality checked by ANA and to 0 otherwise. The second metadata variable, "qual_flag", indicates the reliability of streamflow estimates. It is also

120 provided by ANA and consists of the following quality flags: 0, when there is no description; 1, streamflow resulted from stream stage measurements and the rating curve; 2, streamflow estimated by ANA without stream stage measurements; 3, streamflow values marked as doubtful; and 4, when the stream water level falls outside the range of the stream stage, e.g., when the river ran dry. To summarize the ANA metadata (i.e., q_qual_control_ perc and q_stream_stage_perc; Table 2), 80 % of the 897 gauges had at least 90 % of their data over 1990-2009 checked for inconsistencies (Fig. 1c). The Amazon,

125 São Francisco, and Paraná regions have the lowest frequency of quality controls in Brazil. Furthermore, the streamflow estimates from 64 % of the 897 catchments were derived from stream stage measurements for 90 % of the days over 1990-2009 (Fig. 1d).

**2.2 Meteorological data**

Meteorological daily time series data are provided for 897 catchments (Table 1). These include (i) precipitation from CHIRPS

130 v2.0 (Funk et al., 2015), CPC (NOAA, 2019a), and MSWEP v2.2 (Beck et al., 2019); (ii) potential evapotranspiration from

GLEAM v3.3a (Miralles et al., 2011; Martens et al., 2017); (iii) actual evapotranspiration from GLEAM v3.3a and MGB South America (Siqueira et al., 2018); and (iv) minimum, maximum, and average temperature from CPC (NOAA, 2019b). The datasets were selected because of their high spatial resolution, their full coverage of South America allowing for consistency through all catchments, and because they are commonly used which enables comparisons with other studies. The daily values

135  represent the average of all cells with their centroids intersected by the catchment, of which all cells contribute to the average equally, whether the catchment fully covers them or not. (no weight was applied if a cell is only partially covered by the catchment). However, some catchments do not intersect the centroid of any cell. For those, we computed the daily values as the average of all cells partially covered by the catchment. A significant limitation of the meteorological data is that, because the cell grids of the adopted products have resolutions range from 0.05° (ca. 5.5 km$^2$ at the equator) to 0.5° (ca. 55 km$^2$ at the

140  equator), some catchments are smaller than a single cell. This leads to the assumption that such a meteorological variable is homogeneous in catchments smaller than a single cell, even though this might not always be the case. This limitation has to be kept in mind particularly when using the CPC precipitation data (resolution of 0.5°,: NOAA, 2019), as precipitation is the meteorological variable with the highest spatial heterogeneity amongst those used in CAMELS-BR.

In addition to GLEAM v3.3a, estimates of actual evapotranspiration (ET) were obtained from the MGB model version for

145  South America (Siqueira et al., 2018). The MGB is a conceptual, semi-distributed hydrologic-hydrodynamic model that discretizes the basin (or a set of basins) into irregular unit-catchments and further into hydrological response units by combinations of land use and soil types, where both water and energy balance are computed. The model calculates ET using the Penman-Monteith equation based on CRU meteorological data (i.e., temperature, pressure, radiation, and wind speed) and MSWEP v1.1 precipitation data (Beck et al., 2017b). Surface resistance is adjusted according to the availability of water in the

150  soil that is updated during the water budget. The MGB also computes the evaporation of flooded areas and intercepted water from the canopy with the Penman equation. Regular ET cells of 0.5° resolution were generated by aggregating unit-catchments using their areas as weights.

The long-term water balance is accurate for most catchments, using either the estimated evapotranspiration from GLEAM (Fig. 3a) or MGB (Fig. 3b). Both evapotranspiration data sources indicate that the highest data uncertainties occur in the

155  Amazon and smaller catchments in the Paraná and the Southeastern Atlantic regions since those catchments are further away from the 1:1 line in Fig. 3a-b. The same conclusions are derived from visualizing the runoff coefficient as a function of the humidity index (Fig. 3c). In addition, there are remarkable differences between GLEAM and MGB estimates, where evapotranspiration from GLEAM is substantially higher in the Amazon basin and substantially lower in the Eastern and the Western NE Atlantic regions.

160  **3 Topographic indices**

Even though ANA (2019a) provides estimates of the areas of most gauged Brazilian catchments, the catchment boundaries are not publicly available. Hence, in this study, we used the catchment boundaries provided by Do et al. (2018), who used the

HydroSHEDS 15 arc-sec resolution digital elevation model (DEM) and delineated the catchments with a procedure similar to Lehner (2012) for more than 3,000 gauges in Brazil. For each streamflow gauge, Do et al. (2018) positioned the outlet at the

165 center of all the DEM grid cells within a radius of 5 km from the gauge coordinates indicated by the metadata. They then selected the grid cell (and associated catchment boundaries) leading to the catchment area most similar to the one indicated by ANA (2019a). The main limitation of the procedure of Do et al. (2018) is that catchment boundaries were not manually inspected.

Using those catchment boundaries, we computed four topographic attributes (Table 2), namely gauge elevation, catchment

170 mean elevation, mean slope, and area. The area of the catchments ranged from 10.8 km$^2$ (i.e., in the upper São Francisco hydrographic region) to 4.7 million km$^2$ (i.e., the Amazon basin at Óbidos). Approximately 30 % of the analyzed catchments are smaller than a thousand km$^2$, 43 % are between 1 and 10 thousand km$^2$, and 27 % are larger than 10 thousand km$^2$. The largest basins are in the Amazon and  the Tocantins-Araguaia hydrographic regions (Fig. 4a). Combined with the Paraguay basin, those regions are usually characterized by low elevations (Fig. 4b), flat slopes (Fig. 4c), and large

175 proportions of wetlands (see Sect. 6.2). The smaller catchments are located along the mountain belts on the eastern coast of Brazil, particularly in the southern and southeastern parts of the country. Those are also the catchments with the steepest slopes. Additionally, many catchments with intermediate elevation ranges (i.e., between 500 and 900 m) are in the central part of the country, which comprises the Brazilian highlands. Note that, since we computed the average attribute value (unless otherwise noted) of each catchment, the attributes become less representative as the area of the catchment increases.

180 **4 Climatic indices**

**4.1 Data and methods**

We computed  thirteen climatic indices (Table 3) over the same period (1990 to 2009, except for the asynchronicity index) as the ones in CAMELS (Addor et al., 2017) and CAMELS-Chile (Alvarez-Garreton et al., 2018). The first water year starts on 1st September 1989 and the last one finishes on 31st August 2009. This is to facilitate inter-dataset

185 comparability. We used precipitation data from CHIRPS v2.0 (Funk et al., 2015) to compute the indices, since it has the highest spatial resolution among the three adopted precipitation products (i.e., CHIRPS v2.0, CPC, and MSWEP v2.2) and relies on both remote-sensing and gauge-based data.

The mean precipitation, mean potential evapotranspiration, and the aridity index are considered to capture long-term climatic conditions. The aridity index is the ratio of mean potential evapotranspiration to mean precipitation, which stands as a first-

190 order control on the partitioning of precipitation into streamflow (Budyko et al., 1974; Blöschl et al., 2013). Those indices are complemented by the precipitation seasonality index (p_seasonality, Table 3), which relies on sine curves to approximate the monthly climatology of temperature and precipitation. While, for Brazil, the annual precipitation cycle is captured quite well, a sine curve provides a relatively rough approximation of the temperature cycle, particularly in the center of the country (around the state of Goiás; Berghuijs and Woods, 2016). Hence, in addition to p_seasonality, we extracted the asynchronicity index

195     proposed by Feng et al. (2019), which relies on information theory and has the advantage of being non-parametric (in particular, it does not assume sinusoidality). computed by fitting the seasonal precipitation and temperature using sine curves representing the seasonal amplitude and time of the year when most of the precipitation occurs (Woods, 2009). The indices of extreme climatic conditions include the frequency, duration, and the most common season of high precipitation events and dry days. Dry days are defined as days with precipitation less than 1 mm, so that the index is not compromised by underdetected

200     precipitation events (Haylock and Nicholls, 2000).

**4.2 Spatial variability in climatic indices**

The mean daily precipitation in Brazil is highest in the Amazon and in Southern Brazil, where it usually on average exceeds 5 mm day$^{-1}$ (1825 mm year$^{-1}$) (Fig. 4Fig. 5a). The lowest mean precipitation occurs in Northeastern Brazil, which is also where mean potential evapotranspiration exceeds the mean precipitation (aridity index > 1, Fig. 4Fig. 5b). Northeastern Brazil (in

205     particular, the states of Maranhão, Piauí, Ceará) also has the highest values of asynchronicity index in the country (not shown), which corresponds to Mediterranean climates. The precipitation regime is highly seasonal in most of the country, particularly in the central-west and southeastern Brazil (Fig. 4Fig. 5c). This seasonality is regulated by the South American Monsoon System (Raia and Cavalcanti, 2008; Carvalho et al., 2011), with peaks in the austral summer (Fig. 4Fig. 5f) and several dry months during the austral winter (Fig. 4Fig. 5i). Southern Brazil has a distinct regime, with a uniform precipitation throughout

210     the year caused by a combination of large-scale phenomena and a diversity of sources of atmospheric moisture (Seager et al., 2010; Martinez and Dominguez, 2014). The Amazon basin, which extends into both hemispheres, has contrasting precipitation regimes between the north (with a peak in austral winter) and the south (with a peak in austral summer) related to alternating warming of each hemisphere (Marengo and Espinoza, 2016). This seasonality is substantially diminished downstream in the Amazon.

215     The number of high precipitation and dry days is highest along the catchments on the coast (Fig. 4Fig. 5d and 4g5g), which is also where the smallest catchments are located. Both indices are significantly correlated with catchment area (p-value < 0.001), so a regional analysis of both indices should be carried out with caution since large catchments are located in the Amazon and Tocantins-Araguaia basins. On the other hand, the duration of high precipitation (Fig. 4Fig. 5e) and dry day events (Fig. 4Fig. 5h) do not correlate with catchment area. Their spatial distribution is remarkably similar to the aridity index, except for the

220     Tocantins-Araguaia basin, which has long dry periods but not necessarily long high precipitation events. Summer is the most common season of extreme precipitations in the majority of Brazil, with two main exceptions (Fig. 4Fig. 5f): (i) part of the coast of Northern Brazil; and (ii) Southern Brazil. This is possibly linked to mesoscale convective systems over Southeastern South America (Salio et al., 2007), to sea surface temperature anomalies in the Atlantic ocean (Liebman et al., 2010), and to the El Niño Southern Oscillation phenomenon, as those regions are particularly affected by it (Grimm, 2011; Tedeschi et al.,

225     2013).

**5 Hydrological signatures**

**5.1 Data and methods**

We computed thirteen hydrological signatures (Table 4) that represent a wide range of hydrological information for the water years from 1990 to 2009. The hydrological signatures were computed in the same approach as in CAMELS, CAMEL-CL, and

[revised manuscript text omitted]

820   **Table 4. Hydrological signatures.** **Thresholds for high and low flow frequency and duration were obtained from Clausen and Biggs (2000) and Westerberg and McMillan (2015).**

| Attribute | Description | Units | Data source |
|---|---|---|---|
| q_mean | Mean daily discharge | mm day$^{-1}$ | ANA (2019a) |
| runoff_ratio | Runoff ratio, computed as the ratio of mean daily discharge to mean daily precipitation | - | ANA (2019a) |
| stream_elas | Streamflow precipitation elasticity (i.e., the sensitivity of streamflow to changes in precipitation at the annual timescale, using the mean daily discharge as reference). See equation 7 in Sankarasubramanian et al. (2001), with the last element being $\bar{P}/\bar{Q}$ not $\bar{Q}/\bar{P}$ | - | ANA (2019a) |
| slope_fdc | Slope of the flow duration curve between the log-transformed 33rd and 66th streamflow percentiles | - | ANA (2019a) |
| baseflow_index | Baseflow index, computed as the ratio of mean daily baseflow to mean daily discharge, with the hydrograph separation performed using the Ladson et al. (2013) digital filter | - | ANA (2019a) |
| hfd_mean | Mean half-flow date (i.e., the date on which the cumulative discharge since 1$^{st}$ September reaches half of the annual discharge) | day of the year | ANA (2019a) |
| Q5 | 5% flow quantile (low flow) | mm day$^{-1}$ | ANA (2019a) |
| Q95 | 95% flow quantile (high flow) | mm day$^{-1}$ | ANA (2019a) |
| high_q_freq | Frequency of high-flow days (> 9 times the median daily flow) | days yr$^{-1}$ | ANA (2019a) |
| high_q_dur | Average duration of high-flow events (number of consecutive days > 9 times the median daily flow) | days | ANA (2019a) |
| low_q_freq | Frequency of low-flow days (< 0.2 times the mean daily flow) | days yr$^{-1}$ | ANA (2019a) |
| low_q_dur | Average duration of low-flow events (number of consecutive days < 0.2 times the mean daily flow) | days | ANA (2019a) |
| zero_q_freq | Percentage of days with zero discharge | % | ANA (2019a) |

**Table 5. Land cover characteristics.**

| Attribute | Description | Units | Data source |
|---|---|---|---|
| crop_perc | Percentage covered by croplands | % | ESA GlobCover2009 |
| crop_mosaic_perc | Percentage covered by a mosaic of croplands and natural vegetation | % | ESA GlobCover2009 |
| forest_perc | Percentage covered by broadleaved or needleleaved forests, either evergreen or deciduous | % | ESA GlobCover2009 |
| shrub_perc | Percentage covered by shrublands | % | ESA GlobCover2009 |
| grass_perc | Percentage covered by grasslands or areas with sparse (<15%) vegetation | % | ESA GlobCover2009 |
| barren_perc | Percentage covered by barn areas | % | ESA GlobCover2009 |
| imperv_perc | Percentage covered by artificial surfaces or urban areas | % | ESA GlobCover2009 |
| wet_perc | Percentage covered by water bodies or wetlands | % | ESA GlobCover2009 |
| snow_perc | Percentage covered by permanent snow or ice | % | ESA GlobCover2009 |
| dom_land_cover | Dominant land cover | - | ESA GlobCover2009 |
| dom_land_cover_perc | Percentage covered by the dominant land cover | % | ESA GlobCover2009 |

825

**Table 6. Geologic characteristics.**

| Attribute | Description | Units | Data source |
|---|---|---|---|
| geol_class_1st | Most common geologic class in the catchment | - | GLiM |
| geol_class_1st_perc |  Percentage of the catchment covered by the most common geologic class | % | GLiM |
| geol_class_2nd | Second most common geologic class in the catchment | - | GLiM |
| geol_class_2nd_perc |  Percentage of the catchment covered by the second most common geologic class | % | GLiM |
| carb_rocks_perc |  Percentage of the catchment covered by carbonate sedimentary rocks | % | GLiM |
| geol_porosity | Subsurface porosity of the catchment | - | GLHYMPS v2.0 |
| geol_permeability | Subsurface permeability (log10 scale) of the catchment, extract for each catchment using the geometric mean | m² | GLHYMPS v2.0 |

830 **Table 7. Soil characteristics.**

| Attribute | Description | Units | Data source |
|---|---|---|---|
| sand_perc |  Percentage of sand content of the soil material smaller than 2 mm at a depth of 30 cm | % | SoilGrids250m |
| silt_perc |  Percentage of silt content of the soil material smaller than 2 mm at a depth of 30 cm | % | SoilGrids250m |
| clay_perc |  Percentage of clay content of the soil material smaller than 2 mm at a depth of 30 cm | % | SoilGrids250m |
| org_carbon_content | Soil organic carbon content at a soil depth of 30 cm | g kg$^{-1}$ | SoilGrids250m |
|  bedrock_depth | Depth to bedrock | cm | SoilGrids250m |
| water_table_depth | Median water table depth | cm | Fan et al. (2013) |

**Table 8. Human intervention indices.**

| Attribute | Description | Units | Data source |
|-----------|-------------|-------|-------------|
|  consumptive_use | Total consumptive water use in 2017, normalized by catchment area | mm yr$^{-1}$ | ANA (2019c) |
| consumptive_use_perc | Total consumptive water use in 2017, normalized by mean annual streamflow | % | ANA (2019c) |
|  reservoirs_vol | Total maximum storage capacity of the reservoirs in the catchment | $10^6$ m$^3$ | GRanD v1.3, ONS, and ANA (2018) |
|  regulation_degree | Ratio of total reservoir storage capacity of the catchment to its total annual flow | % | GRanD v1.3, ONS, and ANA (2018) |

835

**Figure 1. (a) South America and the total river discharge data availability  of the 3679 stream gauges included in this study. The black line surrounded by a white line indicates rivers. The dashed line is Brazil's borders. (b) Hydrographic regions of Brazil according to ANA (2019a). (c)  Percentage of streamflow data with quality control checks by ANA of the 897 selected catchments. (d)  Percentage of streamflow data derived from stream stage measurements of the 897 selected catchments. The circles are located at the outlet of the catchments and their sizes are proportional to the sizes of the catchments. The grey line in (c) and (d) indicates the limits of hydrographic regions.**

[Figure]

**Figure 2. Time series with the number of streamflow gauges with at least one measurement for a given year in Brazil.**

[Figure]

845

[Figure]

**Figure 3. Long-term water balance of the 897 selected catchments for the water years 1990-2009. Mean annual evapotranspiration from (a) GLEAM or (b) MGB as a function of the difference between mean annual precipitation (P) from CHIRPS and streamflow (Q). (c) Runoff coefficient as a function of the humidity index. Line 1:1 is shown in black. The symbol size is proportional to the catchment area. The symbol color indicates the hydrographic region of the catchment (from panel d).**

850

**Figure 4. Topographic characteristics of the 897 selected catchments. The size of the circles is proportional to the size of the catchment. The grey line indicates the limits of hydrographic regions.**

[Figure]

**Figure 45. Climatic indices of the 897 selected catchments. The size of the circles is proportional to the size of the catchment. The grey line indicates the limits of hydrographic regions.**

[Figure]

860

**Figure 56. Hydrological signatures of the 897 selected catchments. The black circles are catchments with undefined values. The size of the circles is proportional to the size of the catchment. The grey line indicates the limits of hydrographic regions.**

[Figure]

865    **Figure 67. Land cover characteristics of the 897 selected catchments. The size of the circles is proportional to the size of the catchment. The grey line indicates the limits of hydrographic regions.**

[Figure]

[Figure]

**Figure 9. Soil characteristics of the 897 selected catchments. The size of the circles is proportional to the size of the catchment. The grey line indicates the limits of hydrographic regions.**

[Figure]

875

**Figure 910. Human intervention indices of the 897 selected catchments. The size of the circles is proportional to the size of the catchment. The grey line indicates the limits of hydrographic regions.**

[Figure]